# Bayesian Uncertainty Estimation for Batch Normalized Deep Networks

## Abstract

Deep neural networks have led to a series of breakthroughs, dramatically improving the state-of-the-art in many domains. The techniques driving these advances, however, lack a formal method to account for model uncertainty. While the Bayesian approach to learning provides a solid theoretical framework to handle uncertainty, inference in Bayesian-inspired deep neural networks is difficult. In this paper, we provide a practical approach to Bayesian learning that relies on a regularization technique found in nearly every modern network, batch normalization. We show that training a deep network using batch normalization is equivalent to approximate inference in Bayesian models, and we demonstrate how this finding allows us to make useful estimates of the model uncertainty. Using our approach, it is possible to make meaningful uncertainty estimates using conventional architectures without modifying the network or the training procedure. Our approach is thoroughly validated in a series of empirical experiments on different tasks and using various measures, showing it to outperform baselines on a majority of datasets with strong statistical significance.

## 1 Introduction

Deep learning has dramatically advanced the state of the art in a number of domains, and now surpasses human-level performance for certain tasks such as recognizing the contents of an image (He et al., 2015) and playing Go (Silver et al., 2017). But, despite their unprecedented discriminative power, deep networks are prone to make mistakes. Sometimes, the consequences of mistakes are minor – misidentifying a food dish or a species of flower (Liu et al., 2016) may not be life threatening. But deep networks can already be found in settings where errors carry serious repercussions such as autonomous vehicles (Chen et al., 2016) and high frequency trading. In medicine, we can soon expect automated systems to screen for skin cancer (Esteva et al., 2017), breast cancer (Shen, 2017), and to diagnose biopsies (Djuric et al., 2017). As autonomous systems based on deep learning are increasingly deployed in settings with the potential to cause physical or economic harm, we need to develop a better understanding of when we can be confident in the estimates produced by deep networks, and when we should be less certain.

Standard deep learning techniques used for supervised learning lack methods to account for uncertainty in the model, although sometimes the classification network's output vector is mistakenly understood to represent the model's uncertainty. The lack of a confidence measure can be especially problematic when the network encounters conditions it was not exposed to during training. For example, if a network trained to recognize dog breeds is given an image of a cat, it may predict it to belong to a breed of small dog with high probability. When exposed to data outside of the distribution it was trained on, the network is forced to extrapolate, which can lead to unpredictable behavior. In such cases, if the network can provide information about its uncertainty in addition to its point estimate, disaster may be avoided. This work focuses on estimating such predictive uncertainties in deep networks (Figure 1).

The Bayesian approach provides a solid theoretical framework for modeling uncertainty (Ghahramani, 2015), which has prompted several attempts to extend neural networks (NN) into a Bayesian setting. Most notably, Bayesian neural networks (BNNs) have been studied since the 1990's (Neal, 2012). Although they are simple to formulate, BNNs require substantially more computational resources than their non-Bayesian counterparts, and inference is difficult. Importantly, BNNs do

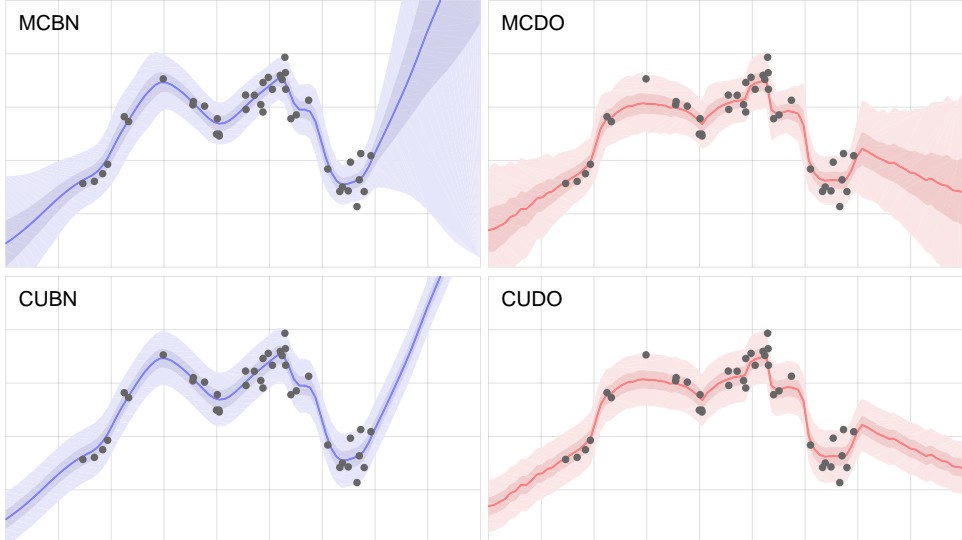

Figure 1: We propose a method to estimate uncertainty in any network using batch normalization (MCBN). Here, we show results on a toy dataset from networks with three hidden layers (30 units per layer). The solid line is the predictive mean of 500 stochastic forward passes. The outer area depicts the model's uncertainty as the 95% CI of the predictive distribution for each $x$ value (inner shaded area is 50% CI). On the right, we show a similar plot using dropout to estimate uncertainty (MCDO) (Gal & Ghahramani, 2015). The bottom row depicts a minimally useful baseline – the same networks but with a constant uncertainty (CUBN, CUDO).

not scale well and struggle to compete with modern deep learning architectures. Recently, Gal & Ghahramani (2015) developed a practical solution to obtain uncertainty estimates by casting *dropout* training in conventional deep networks as an approximate Bayesian model. They showed that *any network* trained with dropout is an approximate Bayesian model, and uncertainty estimates can be obtained by computing the variance on multiple predictions with different dropout masks.

This technique, called *Monte Carlo Dropout* (MCDO), has a very attractive quality: it can be applied to existing NNs without any modification to the architecture or the way the network is trained. Uncertainty estimates come (nearly) for free. However, in recent years dropout has fallen out of favor, limiting MCDO's utility. Google's Inception network, which won ILSVRC in 2014, did not use dropout (Szegedy et al., 2015), nor did the ILSVRC 2015 winner, Microsoft's residual learning network (He et al., 2016). In place of traditional techniques like dropout, most modern networks such as Inception and ResNet have adopted other regularization techniques. In particular, *batch normalization* (BN) has become widespread thanks to its ability to stabilize learning with improved generalization (Ioffe & Szegedy, 2015).

An interesting aspect of BN is that the mini-batch statistics used for training each iteration depend on randomly selected batch members. We exploit this stochasticity and show that training using batch normalization, like dropout, is equivalent to approximate inference in Bayesian models[1]. We demonstrate how this finding allows us to make meaningful estimates of the model uncertainty in a technique we call *Monte Carlo Batch Normalization* (MCBN) (Figure 1). The method we propose makes no simplifying assumptions on the use of batch normalization, and applies to any network using BN as it appears in practical applications.

We validate our approach by empirical experiments on eight standard datasets used for uncertainty estimation. We measure uncertainty quality relative to a baseline of fixed uncertainty, and show that MCBN outperforms the baseline on nearly all datasets with strong statistical significance. We also show that the uncertainty quality of MCBN is on par with that of MCDO. As a practical demonstration of MCBN, we apply our method to estimate segmentation uncertainty using a conventional segmentation network (Badrinarayanan et al., 2015). Finally, as part of our evaluation, we make contributions to the methodology of measuring uncertainty quality by defining performance bounds on existing metrics and proposing a new visualization that provides an intuitive understanding of uncertainty quality.

---

[1]The possibility of using other stochastic regularization techniques is mentioned in Gal (2016).

## 2 RELATED WORK

Bayesian models provide a natural framework for modeling uncertainty, and several approaches have been developed to adapt NNs to Bayesian reasoning. A common approach is to place a prior distribution (often a Gaussian) over each weight. For infinite weights, the resulting model corresponds to a Gaussian process (Neal, 1995), and for a finite number of weights it corresponds to a Bayesian neural network (MacKay, 1992). Although simple to formulate, inference in BNNs is difficult (Gal, 2016). Therefore, focus has shifted to techniques to approximate the posterior distribution, leading to *approximate BNNs*. Methods based on variational inference (VI) typically rely on a fully factorized approximate distribution (Kingma & Welling, 2014; Hinton & Van Camp, 1993) but these methods do not scale easily. To alleviate these difficulties, Graves (2011) proposed a model using sampling methods to estimate a factorized posterior. Another approach, probabilistic backpropagation (PBP), also estimates a factorized posterior based on expectation propagation (Hernández-Lobato & Adams, 2015).

Deep Gaussian Processes (DGPs) formulate GPs as Bayesian models capable of working on large datasets with the aid of a number of strategies to address scaling and complexity requirements (Bui et al., 2016). The authors compare DGP with a number of state-of-the-art approximate BNNs, showing superior performance in terms of RMSE and uncertainty quality[2]. Another recent approach to Bayesian learning, Bayesian hypernetworks, use a neural network to learn a distribution of paramaters over another neural network (Krueger et al., 2017). Although these recent techniques address some of the difficulties with approximate BNNs, they all require modifications to the architecture or the way networks are trained, as well as specialized knowledge from practitioners.

Recently, Gal (2016) showed that a network trained with dropout implicitly performs the VI objective. Therefore *any* network trained with dropout can be treated as an approx. Bayesian model by making multiple predictions as forward passes through the network while sampling different dropout masks for each prediction. An estimate of the posterior can be obtained by computing the mean and variance of the predictions. This technique, referred to here as MCDO, has been empirically demonstrated to be competitive with other approx. BNN methods and DGPs in terms of RMSE and uncertainty quality (Li & Gal, 2017). However, as the name implies, MCDO depends on dropout. While once ubiquitous in training deep learning models, dropout has largely been replaced by batch normalization in modern networks, limiting its usefulness.

## 3 METHOD

The methodology of this work is to pose a deep network trained with batch normalization as a *Bayesian model* in order to obtain uncertainty estimates associated with its predictions. In the following, we briefly introduce Bayesian models and a variational approximation to it using Kullback-Leibler (KL) divergence following Gal & Ghahramani (2015). We continue by showing a batch normalized deep network can be seen as an approximate Bayesian model. Then, by employing theoretical insights as well as empirical analysis, we study the induced prior on the parameters when using batch normalization. Finally, we describe the procedure we use for estimating uncertainty of batch normalized deep networks' output.

### 3.1 BAYESIAN MODELING

We assume a finite training set $\mathbf{D} = \{(\mathbf{x}_i, \mathbf{y}_i)\}_{i=1:N}$ where each $(\mathbf{x}_i, \mathbf{y}_i)$ is a sample-label pair. Using $\mathbf{D}$, we are interested in learning an inference function $f_{\boldsymbol{\omega}}(\mathbf{x}, \mathbf{y})$ with parameters $\boldsymbol{\omega}$. In deterministic models, the estimated label $\hat{\mathbf{y}}$ is obtained as follows:

$$\hat{\mathbf{y}} = \arg\max_{\mathbf{y}} f_{\boldsymbol{\omega}}(\mathbf{x}, \mathbf{y})$$

We assume $f_{\boldsymbol{\omega}}(\mathbf{x}, \mathbf{y}) = p(\mathbf{y}|\mathbf{x}, \boldsymbol{\omega})$ (e.g. in soft-max classifiers), and is normalized to a proper probability distribution. In Bayesian modeling, in contrast to finding a point estimate of the model parameters, the idea is to estimate an (approximate) posterior distribution of the model parameters

---

[2]By uncertainty quality, we refer to predictive probability distributions as measured by PLL and CRPS.

$p(\boldsymbol{\omega}|\mathbf{D})$ to be used for probabilistic prediction:

$$p(\mathbf{y}|\mathbf{x}, \mathbf{D}) = \int f_{\boldsymbol{\omega}}(\mathbf{x}, \mathbf{y}) p(\boldsymbol{\omega}|\mathbf{D}) d\boldsymbol{\omega}$$

The predicted label, $\hat{\mathbf{y}}$, can then be accordingly obtained by sampling $p(\mathbf{y}|\mathbf{x}, \mathbf{D})$ or takings its maxima.

**Variational Approximation**  In *approximate* Bayesian modeling, it is a common approach to learn a parametrized approximating distribution $q_{\boldsymbol{\theta}}(\boldsymbol{\omega})$ that minimizes $\mathrm{KL}(q_{\boldsymbol{\theta}}(\boldsymbol{\omega})||p(\boldsymbol{\omega}|\mathbf{D}))$; the Kullback-Leibler (KL) divergence of posterior w.r.t. its approximation, instead of the true posterior. Minimizing this KL divergence is equivalent to the following minimization while being free of the data term $p(\mathbf{D})$ [3]:

$$\mathcal{L}_{\mathrm{VA}}(\boldsymbol{\theta}) := -\sum_{i=1}^{N} \int q_{\boldsymbol{\theta}}(\boldsymbol{\omega}) \ln f_{\boldsymbol{\omega}}(\mathbf{x}_i, \mathbf{y}_i) d\boldsymbol{\omega} + \mathrm{KL}(q_{\boldsymbol{\theta}}(\boldsymbol{\omega})||p(\boldsymbol{\omega}))$$

Using Monte Carlo integration to approximate the integral with one realized $\hat{\boldsymbol{\omega}}_i$ for each sample $i$ [4], and optimizing over mini-batches of size $M$, the approximated objective becomes:

$$\hat{\mathcal{L}}_{\mathrm{VA}}(\boldsymbol{\theta}) := -\frac{N}{M} \sum_{i=1}^{M} \ln f_{\hat{\boldsymbol{\omega}}_i}(\mathbf{x}_i, \mathbf{y}_i) + \mathrm{KL}(q_{\boldsymbol{\theta}}(\boldsymbol{\omega})||p(\boldsymbol{\omega})) \tag{1}$$

The first term is the data likelihood and the second term is divergence of the model prior w.r.t. the approximated distribution.

We now describe the optimization procedure of a deep network with batch normalization and draw the resemblance to the approximate Bayesian modeling in Eq (1).

## 3.2 BATCH NORMALIZED DEEP NETS AS BAYESIAN MODELING

The inference function of a feed-forward deep network with $L$ layers can be described as:

$$f_{\boldsymbol{\omega}}(\mathbf{x}) = \mathbf{W}^L a(\mathbf{W}^{L-1}...a(\mathbf{W}^2 a(\mathbf{W}^1 \mathbf{x}))$$

where $a(.)$ is an element-wise nonlinearity function and $\mathbf{W}^l$ is the weight vector at layer $l$. Furthermore, we denote the input to layer $l$ as $\mathbf{x}^l$ with $\mathbf{x}^1 = \mathbf{x}$ and we then set $\mathbf{h}^l = \mathbf{W}^l \mathbf{x}^l$. Parenthesized super-index for matrices (e.g. $\mathbf{W}^{(j)}$) and vectors (e.g. $x^{(j)}$) indicates $j$th row and element respectively. Super-index $u$ refers to a specific unit at layer $l$, (e.g. $\mathbf{W}^u = \mathbf{W}^{l,(j)}, h^u = h^{l,(j)}$). [5]

**Batch Normalization**  Each layer of a deep network is constructed by several linear units whose parameters are the rows of the weight matrix $\mathbf{W}$. Batch normalization is a unit-wise operation proposed in Ioffe & Szegedy (2015) to standardize the distribution of each unit's input. It essentially converts a unit's output $h^u$ in the following way:

$$\hat{h}^u = \frac{h^u - \mathbb{E}[h^u]}{\sqrt{\mathrm{Var}[h^u]}}$$

where the expectations are computed over the training set[6]. However, often in deep networks, the weight matrices are optimized using back-propagated errors calculated on mini-batches of data. Therefore, during training, the estimated mean and variance on the mini-batch $\mathbf{B}$ is used, which we denote by $\boldsymbol{\mu}_{\mathbf{B}}$ and $\boldsymbol{\sigma}_{\mathbf{B}}$ respectively. This makes the inference at training time for a sample $\mathbf{x}$ a stochastic process, varying based on other samples in the mini-batch.

---

[3]achieved by constructing the Evidence Lower Bound, called ELBO, and assuming i.i.d. observation noise; details can be found in the appendix sec 6.1.

[4]while a MC integration using a single sample is a weak approximation, in an iterative optimization for $\boldsymbol{\theta}$ several samples will be taken over time.

[5]For a (softmax) classification network, $f_{\boldsymbol{\omega}}(\mathbf{x})$ is a vector with $f_{\boldsymbol{\omega}}(\mathbf{x}, \mathbf{y}) = f_{\boldsymbol{\omega}}(\mathbf{x})^{(\mathbf{y})}$, for regression networks with i.i.d. Gaussian noise we have $f_{\boldsymbol{\omega}}(\mathbf{x}, \mathbf{y}) = \mathcal{N}(f_{\boldsymbol{\omega}}(\mathbf{x}), \tau^{-1}\mathbf{I})$.

[6]It further learns an affine transformation for each unit using parameters $\boldsymbol{\gamma}$ and $\boldsymbol{\beta}$, which we omit in favor of brevity: $\hat{x}_{\mathrm{affine}}^{(j)} = \gamma^{(j)}\hat{x}^{(j)} + \beta^{(j)}$.

**Loss Function and Optimization**  Training deep networks with mini-batch optimization involves a (regularized) risk minimization with the following form:

$$\mathcal{L}_{\text{RR}}(\boldsymbol{\omega}) := \frac{1}{M} \sum_{i=1}^{M} l(\hat{\mathbf{y}}_i, \mathbf{y}_i) + \Omega(\boldsymbol{\omega})$$

Where the first term is the empirical loss on the training data and the second term is a regularization penalty acting as a prior on model parameters $\boldsymbol{\omega}$. If the loss $l$ is cross-entropy for classification or sum-of-squares for regression problems (assuming i.i.d. Gaussian noise on labels), the first term is equivalent to minimizing the negative log-likelihood:

$$\mathcal{L}_{\text{RR}}(\boldsymbol{\omega}) := -\frac{1}{M\tau} \sum_{i=1}^{M} \ln f_{\boldsymbol{\omega}}(\mathbf{x}_i, \mathbf{y}_i) + \Omega(\boldsymbol{\omega}).$$

with $\tau = 1$ for classification. In a batch normalized network the model parameters are $\{\mathbf{W}^{1:L}, \boldsymbol{\gamma}^{1:L}, \boldsymbol{\beta}^{1:L}, \boldsymbol{\mu}_{\mathbf{B}}^{1:L}, \boldsymbol{\sigma}_{\mathbf{B}}^{1:L}\}$. If we decouple the learnable parameters $\boldsymbol{\theta} = \{\mathbf{W}^{1:L}, \boldsymbol{\gamma}^{1:L}, \boldsymbol{\beta}^{1:L}\}$ from the stochastic parameters $\boldsymbol{\omega} = \{\boldsymbol{\mu}_{\mathbf{B}}^{1:L}, \boldsymbol{\sigma}_{\mathbf{B}}^{1:L}\}$, we get the following objective at each step of the mini-batch optimization of a batch normalized network:

$$\mathcal{L}_{\text{RR}}(\boldsymbol{\theta}) := -\frac{1}{M\tau} \sum_{i=1}^{M} \ln f_{\{\boldsymbol{\theta}, \hat{\boldsymbol{\omega}}_i\}}(\mathbf{x}_i, \mathbf{y}_i) + \Omega(\boldsymbol{\theta}) \tag{2}$$

where $\hat{\boldsymbol{\omega}}_i$ is the mean and variances for sample $i$'s mini-batch at a certain training step. Note that while $\hat{\boldsymbol{\omega}}_i$ formally needs to be i.i.d. for each training example, a batch normalized network samples the stochastic parameters once per training step (mini-batch). For a large number of epochs, however, the distribution of sampled batch members for a given training example converges to the i.i.d. case.

Comparing Eq. (1) and Eq. (2) reveals that the optimization objectives are identical, if there exists a prior $p(\boldsymbol{\omega})$ corresponding to $\Omega(\boldsymbol{\theta})$ such that $\frac{\partial}{\partial \theta}\text{KL}(q_{\boldsymbol{\theta}}(\boldsymbol{\omega})||p(\boldsymbol{\omega})) = N\tau\frac{\partial}{\partial \theta}\Omega(\boldsymbol{\theta})$. In a batch normalized network, $q_{\boldsymbol{\theta}}(\boldsymbol{\omega})$ corresponds to the joint distribution of the normalization parameters $\boldsymbol{\mu}_{\mathbf{B}}^{1:L}, \boldsymbol{\sigma}_{\mathbf{B}}^{1:L}$, as implied by the repeated sampling from $\mathbf{D}$ during training. This is an approximation of the true posterior, where we have restricted the posterior to lie within the domain of our parametric network and source of randomness. With that we can use a *pre-trained* batch normalized network to estimate the uncertainty of its prediction using the inherent stochasticity of BN. Before that, we briefly discuss what Bayesian prior is induced in a typical batch normalized network.

### 3.3 PRIOR $p(\boldsymbol{\omega})$

The purpose of $\Omega(\boldsymbol{\theta})$ is to reduce variance in deep networks. L2-regularization, also referred to as weight decay ($\Omega(\boldsymbol{\theta}) = \lambda \sum_{l=1:L} ||W^l||^2$), is a popular technique in deep learning. The induced prior from L2-regularization is studied in Appendix 6.5. Under some approximations as outlined in the Appendix, we find that BN for a deep network with FC layers and ReLU activations induce Gaussian distributions over BN unit's means and standard deviations, centered around the population values given by $\mathbf{D}$ (Eq. (6), details in Appendix 6.3). Factorizing this distribution across all stochastic parameters and assuming Gaussian priors, we find the approximate corresponding priors:

$$p(\mu_{\mathbf{B}}^u) = \mathcal{N}(0, \frac{J_{l-1}x^2}{2N\tau\lambda_l})$$
$$p(\sigma_{\mathbf{B}}^u) = \mathcal{N}(\mu_p, \sigma_p^2)$$

where $J_{l-1}$ is the dimensionality of the layer's inputs and $x$ is the average input over $\mathbf{D}$ for all input units. In the absence of scale and shift transformations from the previous BN layer, it converges towards an exact prior for large training datasets and deep networks (under the assumptions of the factorized distribution). The mean and variance for the BN unit's standard deviation, $\mu_p$ and $\sigma_p^2$, have no relevance for the reconciliation of the optimization objectives of Eq. (1) and (2).

### 3.4 PREDICTIVE UNCERTAINTY IN BATCH NORMALIZED DEEP NETS

In the absence of the true posterior we rely on the approximate posterior to express an approximate predictive distribution:

$$p^*(\mathbf{y}|\mathbf{x}, \mathbf{D}) := \int f_{\boldsymbol{\omega}}(\mathbf{x}, \mathbf{y}) q_{\boldsymbol{\theta}}(\boldsymbol{\omega}) d\boldsymbol{\omega}$$

Following Gal & Ghahramani (2015) we estimate the first and second moment of the predictive distribution empirically (see Appendix 6.4 for details). For regression, the first two moments are:

$$\mathbb{E}_{p^*}[\mathbf{y}] \approx \frac{1}{T} \sum_{i=1}^{T} f_{\hat{\boldsymbol{\omega}}_i}(\mathbf{x})$$

$$\mathrm{Cov}_{p^*}[\mathbf{y}] \approx \tau^{-1}\mathbf{I} + \frac{1}{T} \sum_{i=1}^{T} f_{\hat{\boldsymbol{\omega}}_i}(\mathbf{x})^{\intercal} f_{\hat{\boldsymbol{\omega}}_i}(\mathbf{x}) - \mathbb{E}_{p^*}[\mathbf{y}]^{\intercal}\mathbb{E}_{p^*}[\mathbf{y}]$$

where each $\hat{\boldsymbol{\omega}}_i$ corresponds to sampling the net's stochastic parameters $\boldsymbol{\omega} = \{\boldsymbol{\mu}_{\mathbf{B}}^{1:L}, \boldsymbol{\sigma}_{\mathbf{B}}^{1:L}\}$ the same way as during training. Sampling $\hat{\boldsymbol{\omega}}_i$ therefore involves sampling a batch $\mathbf{B}$ from the *training set* and updating the parameters in the BN units, just as if we were taking a training step with $\mathbf{B}$. Recall that from a VA perspective, training the network amounted to minimizing $\mathrm{KL}(q_{\boldsymbol{\theta}}(\boldsymbol{\omega})\|p(\boldsymbol{\omega}|\mathbf{D}))$ wrt $\boldsymbol{\theta}$. Sampling $\hat{\boldsymbol{\omega}}_i$ from the training set, and keeping the size of $\mathbf{B}$ consistent with the mini-batch size used during training, ensures that $q_{\boldsymbol{\theta}}(\boldsymbol{\omega})$ during inference remains identical to the approximate posterior optimized during training.

After each update of the net's stochastic parameters, we take a forward pass with input $\mathbf{x}$, producing output $f_{\hat{\boldsymbol{\omega}}_i}(\mathbf{x})$. After $T$ such stochastic forward passes, we compute the mean and sample variance of outputs to find the mean $\mathbb{E}_{p^*}[\mathbf{y}]$ and variance $\mathrm{Cov}_{p^*}[\mathbf{y}]$ of the approximate predictive distribution. Note that $\mathrm{Cov}_{p^*}[\mathbf{y}]$ also requires addition of constant variance from observation noise, $\tau^{-1}\mathbf{I}$.

The network is trained just as a regular BN network. The difference is in using the trained network for prediction. Instead of replacing $\boldsymbol{\omega} = \{\boldsymbol{\mu}_{\mathbf{B}}^{1:L}, \boldsymbol{\sigma}_{\mathbf{B}}^{1:L}\}$ with population values from $\mathbf{D}$, we update these parameters stochastically, once for each forward pass .[7]

The form of $p^*$ can be approximated by a Gassuian for each output dimension (for regression). We assume bounded domains for each input dimension, wide layers throughout the network, and a unimodal distribution of weights centered at 0. By the Liapounov CLT condition, the first layer then receives approximately Gaussian inputs (a proof can be found in Lehmann (1999)). Having sampled $\mu_{\mathbf{B}}^u$ and $\sigma_{\mathbf{B}}^u$ from a mini-batch, each BN unit's output is bounded. CLT thereby continues to hold for deeper layers, including $f_{\boldsymbol{\omega}}(\mathbf{x}) = \mathbf{W}^L\mathbf{x}^L$. A similar motivation for a Gaussian approximation of Dropout has been presented by Wang & Manning (2013).

The actual form of $p^*$ is likely to be highly multimodal, as can be seen immediately from $f_{\boldsymbol{\omega}}(\mathbf{x}) = \mathbf{W}^L\mathbf{x}^L$ with elements in $\mathbf{x}^L$ normalized, scaled and shifted differently. Gal & Ghahramani (2015) note the multimodality as well, since MCDO implies a bimodal variational distribution over each weight matrix column.

## 4 EXPERIMENTS AND RESULTS

We assess the uncertainty quality of MCBN quantitatively and qualitatively. Our quantitative analysis relies on eight standard regression datasets, listed in Table 1. Publicly available from the UCI Machine Learning Repository (University of California, 2017) and Delve (Ghahramani, 1996), these datasets have been used to benchmark comparative models in recent related literature (see Hernández-Lobato & Adams (2015), Gal & Ghahramani (2015), Bui et al. (2016) and Li & Gal (2017)). We report results using standard metrics, and also propose useful upper and lower bounds to normalize these metrics for a more meaningful interpretation in Section 4.2.

---

[7]As an alternative to using the training set $\mathbf{D}$ to sample $\hat{\boldsymbol{\omega}}_i$, we could sample from the implied $q_{\boldsymbol{\theta}}(\boldsymbol{\omega})$ as modeled in Appendix 6.3. This would alleviate having to store $\mathbf{D}$ for use during prediction. In our experiments we used $\mathbf{D}$ to sample $\hat{\boldsymbol{\omega}}_i$ however, and leave the evaluation of the modeled $q_{\boldsymbol{\theta}}(\boldsymbol{\omega})$ for future research.

| Dataset name | $N$ | $Q$ | Target Feature |
|---|---|---|---|
| Boston Housing | 506 | 13 | |
| Concrete Compressive Strength | 1,030 | 8 | |
| Energy Efficiency | 768 | 8 | Heating Load |
| Kinematics 8nm | 8,192 | 8 | |
| Power Plant | 9,568 | 4 | |
| Protein Tertiary Structure | 45,730 | 9 | |
| Wine Quality (Red) | 1,599 | 11 | |
| Yacht Hydrodynamics | 308 | 6 | |

Table 1: Properties of the eight regression datasets used to evaluate MCBN. $N$ is the dataset size and $Q$ is the n.o. input features. Only one target feature was used. In cases where the raw datasets contain more than one target feature, the feature used is specified by *target feature*.

Our qualitative results consist of three parts. First, in Figure 1 we demonstrate that MCBN produces reasonable uncertainty bounds on a toy dataset in the style of (Karpathy, 2015). Second, we develop a new visualization of uncertainty quality by plotting test errors sorted by predicted variance in Figure 2. Finally, we apply MCBN to SegNet (Kendall et al., 2015), demonstrating the benefits of MCBN in an existing batch normalized network.

## 4.1 Metrics

We evaluate uncertainty quality based on two metrics, described below: Predictive Log Likelihood (PLL) and Continuous Ranked Probability Score (CRPS). We also propose upper and lower bounds for these metrics which can be used to normalize them and provide a more meaningful interpretation.

**Predictive Log Likelihood (PLL)**   Predictive Log Likelihood is a widely accepted metric for uncertainty quality, used as the main uncertainty quality metric for regression (e.g. (Hernández-Lobato & Adams, 2015), (Gal & Ghahramani, 2015), (Bui et al., 2016) and (Li & Gal, 2017)). A key property is that PLL makes no assumtions about the form of the distribution. The measure is defined for a probabilistic model $f_{\boldsymbol{\omega}}(\mathbf{x})$ and a single observation $(\mathbf{y}_i, \mathbf{x}_i)$ as:

$$\text{PLL}(f_{\boldsymbol{\omega}}(\mathbf{x}), (\mathbf{y}_i, \mathbf{x}_i)) = \log p(\mathbf{y}_i | f_{\boldsymbol{\omega}}(\mathbf{x}_i))$$

where $p(\mathbf{y}_i | f_{\boldsymbol{\omega}}(\mathbf{x}_i))$ is the model's predicted PDF evaluated at $\mathbf{y}_i$, given the input $x_i$. A more detailed description is given in Appendix 6.4. The metric is unbounded and maximized by a perfect prediction (mode at $\mathbf{y}_i$) with no variance. As the predictive mode moves away from $\mathbf{y}_i$, increasing the variance tends to increase PLL (by maximizing probability mass at $\mathbf{y}_i$). While PLL is an elegant measure, it has been criticized for allowing outliers to have an overly negative effect on the score (Selten, 1998).

**Continuous Ranked Probability Score (CRPS)**   Continuous Ranked Probability Score is a less sensitive measure that takes the full predicted PDF into account. A prediction with low variance that is slightly offset from the true observation will receive a higher score form CRPS than PLL. In order for CRPS to be analytically tractable, we need to assume a Gaussian unimodal predictive distribution. CRPS is defined as

$$\text{CRPS}(f_{\boldsymbol{\omega}}(x_i), (y_i, x_i)) = \int_{-\infty}^{\infty} \big(F(y) - \mathbb{1}(y \geq y_i)\big)^2 \mathrm{d}y$$

where $F(y)$ is the predictive CDF, and $\mathbb{1}(y \geq y_i) = 1$ if $y \geq y_i$ and 0 otherwise (for univariate distributions) (Gneiting & Raftery, 2007). CRPS is interpreted as the sum of the squared area between the CDF and 0 where $y < y_i$ and between the CDF and 1 where $y \geq y_i$. A perfect prediction with no variance yields a CRPS of 0; for all other cases the value is larger. CRPS has no upper bound.

## 4.2 Benchmark models and normalized metrics

In order to establish a lower bound on useful performance for uncertainty estimates, we define a baseline that predicts constant variance regardless of input. This benchmark model produces identical point estimates as MCBN, which yield the same predictive means. The variance is set to a fixed value that optimizes CRPS on validation data. This model reflects our best guess of constant

variance on test data - any improvement in uncertainty quality from MCBN would indicate a sensible estimate of uncertainty. We call this model Constant Uncertainty BN (CUBN). Implementing MCDO as a comparative model, we similarly define a baseline for dropout, Constant Uncertainty Dropout (CUDO). The difference in variance modeling between MCBN, CUBN, MCDO and CUDO are visualized in plots of uncertainty bounds on toy data in Figure 1.

For a probabilistic model $f$, an upper bound on uncertainty performance can also be defined for CRPS and PLL. For each observation $(y_i, x_i)$, a value for the predictive variance $T_i$ can be chosen that maximizes PLL or minimizes CRPS[8]. Using CUBN as a lower bound and the optimized CRPS score as the upper bound, uncertainty estimates can be normalized between these bounds (1 indicating optimal performance, and 0 indicating performance on par with fixed uncertainty). We call this normalized measure $\overline{\text{CRPS}} = \frac{\text{CRPS}(f,(y_i,x_i)) - \text{CRPS}(f_{CU},(y_i,x_i))}{\min_T \text{CRPS}(f,(y_i,x_i)) - \text{CRPS}(f_{CU},(y_i,x_i))} \times 100$, and the PLL analogue $\overline{\text{PLL}} = \frac{\text{PLL}(f,(y_i,x_i)) - \text{PLL}(f_{CU},(y_i,x_i))}{\max_T \text{PLL}(f,(y_i,x_i)) - \text{PLL}(f_{CU},(y_i,x_i))} \times 100$. This normalized measure gives an intuitive understanding of how close a Bayesian model is to estimating the *perfect uncertainty* for each prediction.

We also evaluate $\overline{\text{CRPS}}$ and $\overline{\text{PLL}}$ for an adaptation of the authors' implementation of Multiplicative Normalizing Flows (MNF) for variational Bayesian networks (Louizos & Welling, 2017). This is a recent model specialized to allow a more flexible posterior what is achievable by e.g. MCDO's bimodal variational over weight columns. MNF uses auxillary variables on which the posterior is a latent. By applying normalizing flows to the auxillary variable such that it can take on complex distributions, the approximate posterior becomes highly flexible.

### 4.3 TEST SETUP

Our evaluation of MCBN and MCDO is largely comparable to that of Hernández-Lobato & Adams (2015), in that we use similar datasets and metrics. This setup was later also followed by Gal & Ghahramani (2015), where we in comparison implement a different hyperparameter selection, allow for a larger range of dropout rates, and use larger networks with two hidden layers.

With the exception of Protein Tertiary Structure[9], all our models share a similar architecture: two hidden layers with 50 units each, using ReLU activations. Input and output data were normalized during training. Results were averaged over five random splits of $20\%$ test and $80\%$ training and cross-validation (CV) data. For each split, 5-fold CV by grid search with a RMSE minimization objective was used to find training hyperparameters and optimal n.o. epochs. For BN-based models, the hyperparameter grid consisted of a weight decay factor ranging from $0.1$ to $1^{-15}$ by a $\log 10$ scale, and a batch size range from $32$ to $1024$ by a $\log 2$ scale. For DO-based models, the hyperparameter grid consisted of the same weight decay range, and dropout probabilities in $\{0.2, 0.1, 0.05, 0.01, 0.005, 0.001\}$. DO-based models used a batch size of $32$ in all evaluations.

The model with optimal training hyperparameters was used to optimize $\tau$ numerically. This optimization was made in terms of average CV CRPS for MCBN, CUBN, MCDO, and CUDO respectively, before evaluation on the test data.

All estimates for the predictive distribution were obtained by taking 500 stochastic forward passes through the network, throughout training and testing. The implementation was done with TensorFlow. The Adam optimizer was used to train all networks, with a learning rate of 0.001. The extensive part of the experiments (i.e. training and cross validation) was done on Amazon web services using 3000 machine-hours. All code necessary for reproducing both the quantitative and qualitative results is released in an anonymous github repository (https://github.com/iclr-mcbn/mcbn).

### 4.4 TEST RESULTS

A summary of the results measuring uncertainty quality of MCBN, MCDO and MNF are provided in Table 2. Tests are run over eight datasets using 5 random 80-20 splits of the data with 5 different random seeds each split. We report $\overline{\text{CRPS}}$ and $\overline{\text{PLL}}$, expressed as a percentage, which reflects how close the model is to the upper bound. The upper bounds and lower bounds for each metric are de-

---

[8]$T_i$ can be found analytically for PLL, but must be found numerically for CRPS.

[9]Where we used 100 units per hidden layer and 2-fold CV.

| Dataset | $\overline{\text{CRPS}}$ | | | $\overline{\text{PLL}}$ | | |
|---|---|---|---|---|---|---|
| | MCBN | MCDO | MNF | MCBN | MCDO | MNF |
| Boston | **8.50** **** | 3.06 **** | 8.30 **** | **10.49** **** | 5.51 **** | 3.58 *** |
| Concrete | 3.91 **** | 0.93 * | **6.05** **** | -36.36 ** | **10.92** **** | 9.71 **** |
| Energy | **5.75** **** | 1.37 ns | 3.45 ns | **10.89** **** | -14.28 * | 2.62 ns |
| Kin8nm | **2.85** **** | 1.82 **** | 1.01 * | **1.68** *** | -0.26 ns | -0.44 ns |
| Power | 0.24 *** | -0.44 **** | -0.83 *** | 0.33 ** | **3.52** **** | -1.38 **** |
| Protein | **2.66** **** | 0.99 **** | TBU | 2.56 **** | **6.23** **** | TBU |
| Wine (Red) | 0.26 ** | **2.00** **** | TBU | 0.19 * | **2.91** **** | TBU |
| Yacht | -56.39 *** | **21.42** **** | -54.18 **** | 45.58 **** | -41.54 ns | **71.18** **** |

Table 2: Uncertainty quality measured on eight datasets. MCBN, MCDO and MNF are compared over 5 random 80-20 splits of the data with 5 different random seeds each split. Reported values are uncertainty metrics CRPS and PLL normalized to a lower bound of constant variance and upper bound that maximizes the metric. $\overline{\text{CRPS}}$ and $\overline{\text{PLL}}$ are expressed as a percentage, reflecting how close the model is to the upper bound. We check to see if $\overline{\text{CRPS}}$ and $\overline{\text{PLL}}$ significantly exceed the baseline using a one sample t-test (significance level indicated by *'s). Best performer versus their baseline for each dataset and metric is marked by bold. See text for further details.

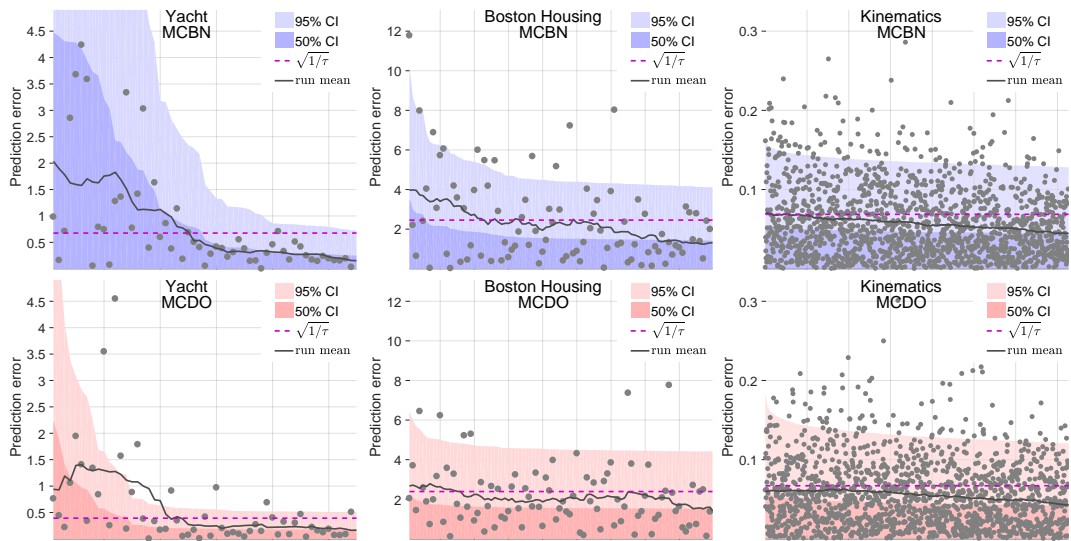

Figure 2: Errors in predictions (gray dots) sorted by estimated uncertainty on select datasets. The shaded areas show MCBN's (blue) and MCDO's (red) model uncertainty (light area 95% CI, dark area 50% CI). Gray dots show absolute prediction errors on the test set, and the gray line depicts a running mean of the errors. The dashed line indicates the optimized constant uncertainty. A correlation between estimated uncertainty (shaded area) and mean error (gray) indicates the uncertainty estimates are meaningful for estimating errors. See Appendix for complete results.

scribed in Section 4.2. We check to see if the reported values of $\overline{\text{CRPS}}$ and $\overline{\text{PLL}}$ significantly exceed the lower bound models (CUBN and CUDO) using a one sample t-test, where the significance level is indicated by *'s. Further details from the experiment are available in Appendix 6.6.

In Figure 2, we provide a novel visualization of uncertainty quality visualization in regression datasets. Errors in the model predictions are sorted by estimated uncertainty. The shaded areas show the model uncertainty and gray dots show absolute prediction errors on the test set. A gray line depicts a running mean of the errors. The dashed line indicates the optimized constant uncertainty. In these plots, we can see a correlation between estimated uncertainty (shaded area) and mean error (gray). This trend indicates that the model uncertainty estimates can recognize samples with larger (or smaller) potential for predictive errors.

Qualitative results for Bayesian SegNet using MCBN was produced by using the main CamVid model in Kendall et al. (2015). The *pre-trained* model was obtained from the online model zoo and was used without modification. 10 instances of mini-batches with size 6 were used to estimate the mean and variance of MCBN. Qualitative results can be found in Figure 3 depicting intuitive

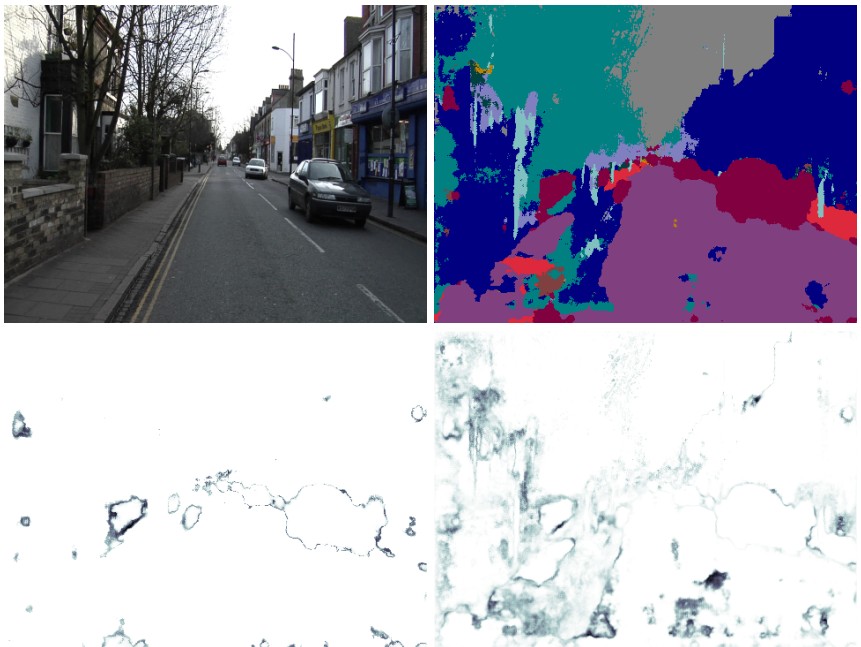

Figure 3: Results applying MCBN to Bayesian SegNet (Kendall et al., 2015). In the upper left, a scene from the CamVid driving scenes dataset. In the upper right, the Bayesian estimated segmentation. In the lower left, estimated uncertainty using MCBN for the car class. In the lower right, the estimated uncertainty of MCBN for all 11 classes.

uncertainty at object boundaries. Quantitative measures on various segmentation datasets can be obtained and is beyond the scope of this work.

We provide additional experimental results in Appendix 6.6. In Tables 3 and 4, we show the mean $\overline{\text{CRPS}}$ and $\overline{\text{PLL}}$ values for MCBN and MCDO. These results indicate that MCBN performs on par with MCDO across several datasets. In Table 6 we provide RMSE results of the MCBN and MCDO networks in comparison with non-stochastic BN and DO networks. These results indicate that the procedure of multiple forward passes in MCBN and MCDO show slight improvements in the predictive accuracy of the network.

## 5 DISCUSSION

The results presented in Table 2 and Appendix 6.6 indicate that MCBN generates meaningful uncertainty estimates which correlate with actual errors in the model's prediction. We show statistically significant improvements over CUBN in the majority of the datasets, both in terms of $\overline{\text{CRPS}}$ and $\overline{\text{PLL}}$. The visualizations in Figure 2 and in Appendix 6.6 show clear correlations between the estimated model uncertainty and actual errors produced by the network. We perform the same experiments using MCDO, and find that MCBN generally performs on par with MCDO. Looking closer, in terms of $\overline{\text{CRPS}}$, MCBN performs better than MCDO in more cases than not. However, care must be used when comparing different models. The learned network parameters are different, leading to different predictive means which can confound direct comparison.

The results on the Yacht Hydrodynamics dataset seem contradictory. The $\overline{\text{CRPS}}$ score for MCBN is extremely negative, while the $\overline{\text{PLL}}$ score is extremely positive. The opposite trend is observed for MCDO. To add to the puzzle, the visualization in Figure 2 depicts an extremely promising uncertainty estimation that models the predictive errors with high fidelity. We hypothesize that this strange behavior is due to the small size of the data set, which only contains 60 test samples, or due to the Gaussian assumption of CRPS. There is also a large variability in the model's accuracy on this dataset, which further confounds the measurements for such limited data.

One might criticize the overall quality of the uncertainty estimates of MCBN and MCDO based on the magnitude of the $\overline{\text{CRPS}}$ and $\overline{\text{PLL}}$ scores in Table 2. The scores rarely exceed 10% improvement over the lower bound. However, we caution that these measures should be taken in context. The upper bound is very difficult to achieve in practice (it is optimized for *each test sample individually*),

and the lower bound is a quite reasonable estimate for uncertainty. We have further compared against the recent work of Louizos & Welling (2017), and find comparable results to their MNF-based variational technique specifically targeted to increase the flexibility of the approximate posterior.

Our approximation of the implied prior in Appendix 6.5 also provides a new interpretation of the empirical evidence that significantly lower $\lambda$ should be used in batch normalized networks (Ioffe & Szegedy, 2015). From a VA perspective, too strong a regularization for a given dataset size could be seen as constraining the prior distribution of BN units' means, effectively narrowing the approximate posterior.

In this work, we have shown that training a deep network using batch normalization is equivalent to approximate inference in Bayesian models. Using our approach, it is possible to make meaningful uncertainty estimates using conventional architectures without modifying the network or the training procedure. We show evidence that the uncertainty estimates from MCBN correlate with actual errors in the model's prediction, and are useful for practical tasks such as regression or semantic image segmentation. Our experiments show that MCBN yields an improvement over the baseline of optimized constant uncertainty on par with MCDO and MNF. Finally, we make contributions to the evaluation of uncertainty quality by suggesting new evaluation metrics based on useful baselines and upper bounds, and proposing a new visualization tool which gives an intuitive visual explanation of uncertainty quality. Finally, it should be noted that, over the past few years, batch normalization has become an integral part of most-if-not-all cutting edge deep networks which signifies the relevance of our work for estimating model uncertainty.

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

## 6 APPENDIX

### 6.1 VARIATIONAL APPROXIMATION

Assume we were to come up with a faimly of distributions parametrised by $\boldsymbol{\theta}$ in order to approximate the posterior, $q_{\boldsymbol{\theta}}(\boldsymbol{\omega})$. Our goal is to set $\boldsymbol{\theta}$ such that $q_{\boldsymbol{\theta}}(\boldsymbol{\omega})$ is as similar to $p(\boldsymbol{\omega}|\mathbf{D})$ as possible.

One strategy is to minimizing $\text{KL}(q_{\boldsymbol{\theta}}(\boldsymbol{\omega})||p(\boldsymbol{\omega}|\mathbf{D}))$, the KL divergence of $p(\boldsymbol{\omega}|\mathbf{D})$ wrt $q_{\boldsymbol{\theta}}(\boldsymbol{\omega})$. Minimizing $\text{KL}(q_{\boldsymbol{\theta}}(\boldsymbol{\omega})||p(\boldsymbol{\omega}|\mathbf{D}))$ is equivalent to maximizing the ELBO:

$$\int_{\boldsymbol{\omega}} q_{\boldsymbol{\theta}}(\boldsymbol{\omega}) \ln p(\mathbf{Y}|\mathbf{X}, \boldsymbol{\omega}) \mathrm{d}\boldsymbol{\omega} - \text{KL}(q_{\boldsymbol{\theta}}(\boldsymbol{\omega})||p(\boldsymbol{\omega}))$$

Assuming i.i.d. observation noise, this is equivalent to minimizing:

$$\mathcal{L}_{\text{VA}}(\boldsymbol{\theta}) := -\sum_{n=1}^{N} \int q_{\boldsymbol{\theta}}(\boldsymbol{\omega}) \ln p(\mathbf{y}_i|f_{\boldsymbol{\omega}}(\mathbf{x}_i)) \mathrm{d}\boldsymbol{\omega} + \text{KL}(q_{\boldsymbol{\theta}}(\boldsymbol{\omega})||p(\boldsymbol{\omega}))$$

Instead of making the optimization on the full training set, we can use a subsampling (yielding an unbiased estimate of $\mathcal{L}_{\text{VA}}(\boldsymbol{\theta})$) for iterative optimization (as in mini-batch optimization):

$$\hat{\mathcal{L}}_{\text{VA}}(\boldsymbol{\theta}) := -\frac{N}{M} \sum_{i \in B} \int_{\boldsymbol{\omega}} q_{\boldsymbol{\theta}}(\boldsymbol{\omega}) \ln p(\mathbf{y}_i|f_{\boldsymbol{\omega}}(\mathbf{x}_i)) \mathrm{d}\boldsymbol{\omega} + \text{KL}(q_{\boldsymbol{\theta}}(\boldsymbol{\omega})||p(\boldsymbol{\omega}))$$

We now make a reparametrisation: set $\boldsymbol{\omega} = g(\boldsymbol{\theta}, \epsilon)$ where $\epsilon$ is a RV. The function $g$ and the distribution of $\epsilon$ must be such that $p(g(\boldsymbol{\theta}, \epsilon)) = q_{\boldsymbol{\theta}}(\boldsymbol{\omega})$. Assume $q_{\boldsymbol{\theta}}(\boldsymbol{\omega})$ can be written $\int_{\epsilon} q_{\boldsymbol{\theta}}(\boldsymbol{\omega}|\epsilon)p(\epsilon)\mathrm{d}\epsilon$ where $q_{\boldsymbol{\theta}}(\boldsymbol{\omega}|\epsilon) = \delta(\boldsymbol{\omega} - g(\boldsymbol{\theta}, \epsilon))$. Using this reparametrisation we get:

$$\hat{\mathcal{L}}_{\text{VA}}(\boldsymbol{\theta}) = -\frac{N}{M} \sum_{i \in B} \int_{\epsilon} p(\epsilon) \ln p(\mathbf{y}_i|f_{g(\boldsymbol{\theta}, \epsilon)}(\mathbf{x}_i)) \mathrm{d}\epsilon + \text{KL}(q_{\boldsymbol{\theta}}(\boldsymbol{\omega})||p(\boldsymbol{\omega}))$$

### 6.2 KL DIVERGENCE OF FACTORIZED GAUSSIANS

If $q_{\boldsymbol{\theta}}(\boldsymbol{\omega})$ and $p(\boldsymbol{\omega})$ factorize over all stochastic parameters:

$$
\begin{aligned}
\text{KL}(q_{\boldsymbol{\theta}}(\boldsymbol{\omega})||p(\boldsymbol{\omega})) &= -\int_{\boldsymbol{\omega}} \prod_i [q_{\boldsymbol{\theta}}(\omega_i)] \ln \frac{\prod_i p(\omega_i)}{\prod_i q_{\boldsymbol{\theta}}(\omega_i)} \mathrm{d}\boldsymbol{\omega} \\
&= -\int_{\boldsymbol{\omega}} \prod_i [q_{\boldsymbol{\theta}}(\omega_i)] \sum_i \left[ \ln \frac{p(\omega_i)}{q_{\boldsymbol{\theta}}(\omega_i)} \right] \prod_i \mathrm{d}\omega_i \\
&= \sum_j \left[ -\int_{\boldsymbol{\omega}} \prod_i [q_{\boldsymbol{\theta}}(\omega_i)] \ln \frac{p(\omega_j)}{q_{\boldsymbol{\theta}}(\omega_j)} \prod_i \mathrm{d}\omega_i \right] \\
&= \sum_j \left[ -\int_{\omega_j} q_{\boldsymbol{\theta}}(\omega_j) \ln \frac{p(\omega_j)}{q_{\boldsymbol{\theta}}(\omega_j)} \mathrm{d}\omega_j \int_{\omega_{i \neq j}} \prod_{i \neq j} q_{\boldsymbol{\theta}}(\omega_i) \mathrm{d}\omega_i \right] \\
&= \sum_i -\int_{\omega_i} q_{\boldsymbol{\theta}}(\omega_i) \ln \frac{p(\omega_i)}{q_{\boldsymbol{\theta}}(\omega_i)} \mathrm{d}\omega_i \\
&= \sum_i \text{KL}(q_{\boldsymbol{\theta}}(\omega_i)||p(\omega_i))
\end{aligned}
\tag{3}
$$

such that $\text{KL}(q_{\boldsymbol{\theta}}(\boldsymbol{\omega})||p(\boldsymbol{\omega}))$ is the sum of the KL divergence terms for the individual stochastic parameters $\omega_i$. If the factorized distributions are Gaussians, where $q_{\boldsymbol{\theta}}(\omega_i) = \mathcal{N}(\mu_q, \sigma_q^2)$ and $p(\omega_i) =$

$\mathcal{N}(\mu_p, \sigma_p^2)$ we get:

$$
\begin{aligned}
\mathrm{KL}(q_{\boldsymbol{\theta}}(\omega_i)||p(\omega_i)) &= \int_{\omega_i} q_{\boldsymbol{\theta}}(\omega_i) \ln \frac{q_{\boldsymbol{\theta}}(\omega_i)}{p(\omega_i)} \mathrm{d}\omega_i \\
&= - H(q_{\boldsymbol{\theta}}(\omega_i)) - \int_{\omega_i} q_{\boldsymbol{\theta}}(\omega_i) \ln p(\omega_i) \mathrm{d}\omega_i \\
&= - \frac{1}{2}(1 + \ln(2\pi\sigma_q^2)) \\
&\quad - \int_{\omega_i} q_{\boldsymbol{\theta}}(\omega_i) \ln \frac{1}{(2\pi\sigma_p^2)^{1/2}} \exp\left\{ - \frac{(\omega_i - \mu_p)^2}{2\sigma_p^2} \right\} \mathrm{d}\omega_i \\
&= - \frac{1}{2}(1 + \ln(2\pi\sigma_q^2)) \\
&\quad + \frac{1}{2} \ln(2\pi\sigma_p^2) + \frac{\mathbb{E}_q[\omega_i^2] - 2\mu_p \mathbb{E}_q[\omega_i] + \mu_p^2}{2\sigma_p^2} \\
&= \ln \frac{\sigma_p}{\sigma_q} + \frac{\sigma_q^2 + (\mu_q - \mu_p)^2}{2\sigma_p^2} - \frac{1}{2}
\end{aligned}
\tag{4}
$$

for each KL divergence term. Here $H(q_{\boldsymbol{\theta}}(\omega_i)) = \frac{1}{2}(1 + \ln(2\pi\sigma_q^2))$ is the differential entropy of $q_{\boldsymbol{\theta}}(\omega_i)$.

## 6.3 DISTRIBUTION OF $\mu_{\mathbf{B}}^u, \sigma_{\mathbf{B}}^u$

Here we approximate the distribution of mean and standard deviation of a mini-batch, separately to two Gaussians. For the mean we get:

$$
\mu_{\mathbf{B}} = \frac{\Sigma_{\mathrm{m}=1}^{\mathrm{M}} \mathbf{W}^{(j)} \mathbf{x}_{\mathrm{m}}}{\mathbf{M}}
$$

where $\mathbf{x}_{\mathrm{m}}$ are the examples in the sampled batch. We will assume these are sampled i.i.d.[10]. Samples of the random variable $\mathbf{W}^{(j)} \mathbf{x}_{\mathrm{m}}$ are then i.i.d.. Then by central limit theorem (CLT) the following holds for sufficiently large M (often $\geq 30$):

$$
\mu_{\mathbf{B}} \sim \mathcal{N}(\mu, \frac{\sigma^2}{\mathbf{M}})
$$

For standard deviation:

$$
\sigma_{\mathbf{B}} = \sqrt{\frac{\Sigma_{\mathrm{m}=1}^{\mathrm{M}} (\mathbf{W}^{(j)} \mathbf{x}_{\mathrm{m}} - \mu_{\mathbf{B}})^2}{M}}
$$

Then

$$
\sqrt{M}(\sigma_{\mathbf{B}} - \sigma) = \sqrt{M}\left( \sqrt{\frac{\Sigma_{\mathrm{m}=1}^{\mathrm{M}} (\mathbf{W}^{(j)} \mathbf{x}_{\mathrm{m}} - \mu_{\mathbf{B}})^2}{M}} - \sqrt{\sigma^2} \right)
$$

We want to rewrite $\sqrt{\frac{\Sigma_{\mathrm{m}=1}^{\mathrm{M}} (\mathbf{W}^{(j)} \mathbf{x}_{\mathrm{m}} - \mu_{\mathbf{B}})^2}{M}}$. We take a Taylor expansion of $f(x) = \sqrt{x}$ around $a = \sigma^2$. With $x = \frac{\Sigma_{\mathrm{m}=1}^{\mathrm{M}} (\mathbf{W}^{(j)} \mathbf{x}_{\mathrm{m}} - \mu_{\mathbf{B}})^2}{M}$:

$$
\sqrt{x} = \sqrt{\sigma^2} + \frac{1}{2\sqrt{\sigma^2}}(x - \sigma^2) + \mathcal{O}[(x - \sigma^2)^2]
$$

---

[10]Although in practice with deep learning, mini-batches are sampled without replacement, stochastic gradient descent samples with replacement in its standard form.

so

$$
\sqrt{M}(\sigma_{\mathrm{B}} - \sigma) = \sqrt{M}\left(\frac{1}{2\sqrt{\sigma^2}}\left(\frac{\Sigma_{m=1}^{M}(\mathbf{W}^{(j)}\mathbf{x}_m - \mu_{\mathrm{B}})^2}{M} - \sigma^2\right) + \right.
$$
$$
\left. \mathcal{O}\left[\left(\frac{\Sigma_{m=1}^{M}(\mathbf{W}^{(j)}\mathbf{x}_m - \mu_{\mathrm{B}})^2}{M} - \sigma^2\right)^2\right]\right)
$$
$$
= \frac{\sqrt{M}}{2\sigma}\left(\frac{1}{M}\Sigma_{m=1}^{M}(\mathbf{W}^{(j)}\mathbf{x}_m - \mu_{\mathrm{B}})^2 - \sigma^2\right) +
$$
$$
\mathcal{O}\left[\sqrt{M}\left(\frac{\Sigma_{m=1}^{M}(\mathbf{W}^{(j)}\mathbf{x}_m - \mu_{\mathrm{B}})^2}{M} - \sigma^2\right)^2\right]
$$
$$
= \frac{1}{2\sigma\sqrt{M}}\left(\Sigma_{m=1}^{M}(\mathbf{W}^{(j)}\mathbf{x}_m - \mu_{\mathrm{B}})^2 - M\sigma^2\right) +
$$
$$
\mathcal{O}\left[\sqrt{M}\left(\frac{\Sigma_{m=1}^{M}(\mathbf{W}^{(j)}\mathbf{x}_m - \mu_{\mathrm{B}})^2}{M} - \sigma^2\right)^2\right]
$$

consider $\Sigma_{m=1}^{M}(\mathbf{W}^{(j)}\mathbf{x}_m - \mu_{\mathrm{B}})^2$. We know that $E[\mathbf{W}^{(j)}\mathbf{x}_m] = \mu$ and write

$$
\Sigma_{m=1}^{M}(\mathbf{W}^{(j)}\mathbf{x}_m - \mu_{\mathrm{B}})^2
$$
$$
= \Sigma_{m=1}^{M}((\mathbf{W}^{(j)}\mathbf{x}_m - \mu) - (\mu_{\mathrm{B}} - \mu))^2
$$
$$
= \Sigma_{m=1}^{M}((\mathbf{W}^{(j)}\mathbf{x}_m - \mu)^2 + (\mu_{\mathrm{B}} - \mu)^2 - 2(\mathbf{W}^{(j)}\mathbf{x}_m - \mu)(\mu_{\mathrm{B}} - \mu))
$$
$$
= \Sigma_{m=1}^{M}(\mathbf{W}^{(j)}\mathbf{x}_m - \mu)^2 + M(\mu_{\mathrm{B}} - \mu)^2 - 2(\mu_{\mathrm{B}} - \mu)\Sigma_{m=1}^{M}(\mathbf{W}^{(j)}\mathbf{x}_m - \mu)
$$
$$
= \Sigma_{m=1}^{M}(\mathbf{W}^{(j)}\mathbf{x}_m - \mu)^2 - M(\mu_{\mathrm{B}} - \mu)^2
$$
$$
= \Sigma_{m=1}^{M}((\mathbf{W}^{(j)}\mathbf{x}_m - \mu)^2 - (\mu_{\mathrm{B}} - \mu)^2)
$$

then

$$\sqrt{M}(\sigma_B - \sigma) = \frac{1}{2\sigma\sqrt{M}}\left(\Sigma_{m=1}^M((\mathbf{W}^{(j)}\mathbf{x}_m - \mu)^2 - (\mu_B - \mu)^2) - M\sigma^2\right)+$$

$$\mathcal{O}\left[\sqrt{M}\left(\frac{\Sigma_{m=1}^M(\mathbf{W}^{(j)}\mathbf{x}_m - \mu_B)^2}{M} - \sigma^2\right)^2\right]$$

$$= \frac{1}{2\sigma\sqrt{M}}\left(\Sigma_{m=1}^M(\mathbf{W}^{(j)}\mathbf{x}_m - \mu)^2 - \Sigma_{m=1}^M(\mu_B - \mu)^2 - M\sigma^2\right)+$$

$$\mathcal{O}\left[\sqrt{M}\left(\frac{\Sigma_{m=1}^M(\mathbf{W}^{(j)}\mathbf{x}_m - \mu_B)^2}{M} - \sigma^2\right)^2\right]$$

$$= \frac{1}{2\sigma\sqrt{M}}\left(\Sigma_{m=1}^M((\mathbf{W}^{(j)}\mathbf{x}_m - \mu)^2 - \sigma^2) - \Sigma_{m=1}^M(\mu_B - \mu)^2\right)+$$

$$\mathcal{O}\left[\sqrt{M}\left(\frac{\Sigma_{m=1}^M(\mathbf{W}^{(j)}\mathbf{x}_m - \mu_B)^2}{M} - \sigma^2\right)^2\right]$$

$$= \frac{1}{2\sigma\sqrt{M}}\Sigma_{m=1}^M((\mathbf{W}^{(j)}\mathbf{x}_m - \mu)^2 - \sigma^2)$$

$$- \frac{1}{2\sigma\sqrt{M}}\Sigma_{m=1}^M(\mu_B - \mu)^2$$

$$+ \mathcal{O}\left[\sqrt{M}\left(\frac{\Sigma_{m=1}^M(\mathbf{W}^{(j)}\mathbf{x}_m - \mu_B)^2}{M} - \sigma^2\right)^2\right]$$

$$= \underbrace{\frac{1}{2\sigma\sqrt{M}}\Sigma_{m=1}^M((\mathbf{W}^{(j)}\mathbf{x}_m - \mu)^2 - \sigma^2)}_{\text{term A}}$$

$$\underbrace{- \frac{\sqrt{M}}{2\sigma}(\mu_B - \mu)^2}_{\text{term B}}$$

$$\underbrace{+ \mathcal{O}\left[\sqrt{M}\left(\frac{\Sigma_{m=1}^M(\mathbf{W}^{(j)}\mathbf{x}_m - \mu_B)^2}{M} - \sigma^2\right)^2\right]}_{\text{term C}}$$

We go through each term in turn

**Term A**
We have

$$\text{Term A} = \frac{1}{2\sigma\sqrt{M}}\Sigma_{m=1}^M((\mathbf{W}^{(j)}\mathbf{x}_m - \mu)^2 - \sigma^2)$$

where $\Sigma_{m=1}^M(\mathbf{W}^{(j)}\mathbf{x}_m - \mu)^2$ is the sum of $M$ RVs $(\mathbf{W}^{(j)}\mathbf{x}_m - \mu)^2$. Note that since $E[\mathbf{W}^{(j)}\mathbf{x}_m] = \mu$ it holds that $E[(\mathbf{W}^{(j)}\mathbf{x}_m - \mu)^2] = \sigma^2$. Since $(\mathbf{W}^{(j)}\mathbf{x}_m - \mu)^2$ is sampled approximately iid (by assumptions above), for large enough M by CLT it holds approximately that

$$\Sigma_{m=1}^M(\mathbf{W}^{(j)}\mathbf{x}_m - \mu)^2 \sim \mathcal{N}(M\sigma^2, M\text{Var}((\mathbf{W}^{(j)}\mathbf{x}_m - \mu)^2))$$

where

$$\text{Var}((\mathbf{W}^{(j)}\mathbf{x}_m - \mu)^2) = E[(\mathbf{W}^{(j)}\mathbf{x}_m - \mu)^{2*2}] - E[(\mathbf{W}^{(j)}\mathbf{x}_m - \mu)^2]^2$$
$$= E[(\mathbf{W}^{(j)}\mathbf{x}_m - \mu)^4] - \sigma^4$$

Then

$$\Sigma_{m=1}^M((\mathbf{W}^{(j)}\mathbf{x}_m - \mu)^2 - \sigma^2) \sim \mathcal{N}(0, M * E[(\mathbf{W}^{(j)}\mathbf{x}_m - \mu)^4] - M\sigma^4)$$

so

$$\text{Term A} \sim \mathcal{N}(0, \frac{E[(\mathbf{W}^{(j)}\mathbf{x}_m - \mu)^4] - \sigma^4}{4\sigma^2})$$

**Term B**
We have

$$\text{Term B} = \frac{\sqrt{M}}{2\sigma}(\mu_\text{B} - \mu)^2 = \frac{1}{2\sigma}\sqrt{M}(\mu_\text{B} - \mu)(\mu_\text{B} - \mu)$$

Consider $(\mu_\text{B} - \mu)$. As $\mu_\text{B} \xrightarrow{p} \mu$ when $M \to \infty$ we have $\mu_\text{B} - \mu \xrightarrow{p} 0$. We also have

$$\sqrt{M}(\mu_\text{B} - \mu) = \frac{\Sigma_{\text{m}=1}^\text{M} \mathbf{W}^{(j)}\mathbf{x}_\text{m}}{\sqrt{M}} - \sqrt{M}\mu$$

which by CLT is approximately Gaussian for large $M$. We can then make use of the Cramer-Slutzky Theorem, which states that if $(X_n)_{n \geq 1}$ and $(Y_n)_{n \geq 1}$ are two sequences such that $X_n \xrightarrow{d} X$ and $Y_n \xrightarrow{p} a$ as $n \to \infty$ where $a$ is a constant, then as $n \to \infty$, it holds that $X_n * Y_n \xrightarrow{d} X * a$. Thus, Term B is approximately 0 for large M.

**Term C**
We have

$$\text{Term C} = \mathcal{O}\left[\sqrt{M}\Big(\frac{\Sigma_{\text{m}=1}^\text{M}(\mathbf{W}^{(j)}\mathbf{x}_\text{m} - \mu_\text{B})^2}{M} - \sigma^2\Big)^2\right]$$

Since $E[(\mathbf{W}^{(j)}\mathbf{x}_\text{m} - \mu)^2] = \sigma^2$ we can make the same use of Cramer-Slutzky as for *Term B*, such that Term C is approximately 0 for large M.

**Finalizing the distribution**
We have approximately

$$\sqrt{M}(\sigma_\text{B} - \sigma) \sim \mathcal{N}(0, \frac{E[(\mathbf{W}^{(j)}\mathbf{x}_\text{m} - \mu)^4] - \sigma^4}{4\sigma^2})$$

so

$$\sigma_\text{B} \sim \mathcal{N}(\sigma, \frac{E[(\mathbf{W}^{(j)}\mathbf{x}_\text{m} - \mu)^4] - \sigma^4}{4\sigma^2 M})$$

### 6.4    PREDICTIVE DISTRIBUTION PROPERTIES

This section provides derivations of properties of the predictive distribution $p^*(\mathbf{y}|\mathbf{x}, \mathbf{D})$ in section 3.4, following Gal (2016). We first find the approximate predictive mean and variance for the approximate predictive distribution, then show how to estimate the predictive log likelihood, a measure of uncertainty quality used in the evaluation 4.

**Predictive mean**    Assuming Gaussian iid noise defined by model precision $\tau$, i.e. $f_{\boldsymbol{\omega}}(\mathbf{x}, \mathbf{y}) = p(\mathbf{y}|f_{\boldsymbol{\omega}}(\mathbf{x})) = \mathcal{N}(\mathbf{y}; f_{\boldsymbol{\omega}}(\mathbf{x}), \tau^{-1}\mathbf{I})$:

$$\mathbb{E}_{p^*}[\mathbf{y}] = \int \mathbf{y} p^*(\mathbf{y}|\mathbf{x}, \mathbf{D}) d\mathbf{y}$$

$$= \int_\mathbf{y} \mathbf{y}\Big(\int_{\boldsymbol{\omega}} f_{\boldsymbol{\omega}}(\mathbf{x}, \mathbf{y}) q_{\boldsymbol{\theta}}(\boldsymbol{\omega}) d\boldsymbol{\omega}\Big) d\mathbf{y}$$

$$= \int_\mathbf{y} \mathbf{y}\Big(\int_{\boldsymbol{\omega}} \mathcal{N}(\mathbf{y}; f_{\boldsymbol{\omega}}(\mathbf{x}), \tau^{-1}\mathbf{I}) q_{\boldsymbol{\theta}}(\boldsymbol{\omega}) d\boldsymbol{\omega}\Big) d\mathbf{y}$$

$$= \int_{\boldsymbol{\omega}} \Big(\int_\mathbf{y} \mathbf{y}\mathcal{N}(\mathbf{y}; f_{\boldsymbol{\omega}}(\mathbf{x}), \tau^{-1}\mathbf{I}) d\mathbf{y}\Big) q_{\boldsymbol{\theta}}(\boldsymbol{\omega}) d\boldsymbol{\omega}$$

$$= \int_{\boldsymbol{\omega}} f_{\boldsymbol{\omega}}(\mathbf{x}) q_{\boldsymbol{\theta}}(\boldsymbol{\omega}) d\boldsymbol{\omega}$$

$$\approx \frac{1}{T}\sum_{i=1}^T f_{\hat{\boldsymbol{\omega}}_i}(\mathbf{x})$$

where we take the MC Integral with $T$ samples of $\boldsymbol{\omega}$ for the approximation in the final step.

**Predictive variance**   Our goal is to estimate:

$$\text{Cov}_{p^*}[\mathbf{y}] = \mathbb{E}_{p^*}[\mathbf{y}^\mathsf{T}\mathbf{y}] - \mathbb{E}_{p^*}[\mathbf{y}]^\mathsf{T}\mathbb{E}_{p^*}[\mathbf{y}]$$

We find that:

$$
\begin{aligned}
\mathbb{E}_{p^*}[\mathbf{y}^\mathsf{T}\mathbf{y}] &= \int_{\mathbf{y}} \mathbf{y}^\mathsf{T}\mathbf{y}\, p^*(\mathbf{y}|\mathbf{x},\mathbf{D})\mathrm{d}\mathbf{y} \\
&= \int_{\mathbf{y}} \mathbf{y}^\mathsf{T}\mathbf{y}\Big( \int_{\boldsymbol{\omega}} f_{\boldsymbol{\omega}}(\mathbf{x},\mathbf{y})q_{\boldsymbol{\theta}}(\boldsymbol{\omega})d\boldsymbol{\omega}\Big)\mathrm{d}\mathbf{y} \\
&= \int_{\boldsymbol{\omega}} \Big( \int_{\mathbf{y}} \mathbf{y}^\mathsf{T}\mathbf{y} f_{\boldsymbol{\omega}}(\mathbf{x},\mathbf{y})\mathrm{d}\mathbf{y}\Big)q_{\boldsymbol{\theta}}(\boldsymbol{\omega})\mathrm{d}\boldsymbol{\omega} \\
&= \int_{\boldsymbol{\omega}} \Big( \text{Cov}_{f_{\boldsymbol{\omega}}(\mathbf{x},\mathbf{y})}(\mathbf{y}) + \mathbb{E}_{f_{\boldsymbol{\omega}}(\mathbf{x},\mathbf{y})}[\mathbf{y}]^\mathsf{T}\mathbb{E}_{f_{\boldsymbol{\omega}}(\mathbf{x},\mathbf{y})}[\mathbf{y}]\Big)q_{\boldsymbol{\theta}}(\boldsymbol{\omega})\mathrm{d}\boldsymbol{\omega} \\
&= \int_{\boldsymbol{\omega}} \Big( \tau^{-1}\mathbf{I} + f_{\boldsymbol{\omega}}(\mathbf{x})^\mathsf{T} f_{\boldsymbol{\omega}}(\mathbf{x})\Big)q_{\boldsymbol{\theta}}(\boldsymbol{\omega})\mathrm{d}\boldsymbol{\omega} \\
&= \tau^{-1}\mathbf{I} + E_{q_{\boldsymbol{\theta}}(\boldsymbol{\omega})}[f_{\boldsymbol{\omega}}(\mathbf{x})^\mathsf{T} f_{\boldsymbol{\omega}}(\mathbf{x})] \\
&\approx \tau^{-1}\mathbf{I} + \frac{1}{T}\sum_{i=1}^{T} f_{\hat{\boldsymbol{\omega}}_i}(\mathbf{x})^\mathsf{T} f_{\hat{\boldsymbol{\omega}}_i}(\mathbf{x})
\end{aligned}
$$

where we use MC integration with $T$ samples for the final step. The predictive covariance matrix is given by:

$$\text{Cov}_{p^*}[\mathbf{y}] \approx \tau^{-1}\mathbf{I} + \frac{1}{T}\sum_{i=1}^{T} f_{\hat{\boldsymbol{\omega}}_i}(\mathbf{x})^\mathsf{T} f_{\hat{\boldsymbol{\omega}}_i}(\mathbf{x}) - \mathbb{E}_{p^*}[\mathbf{y}]^\mathsf{T}\mathbb{E}_{p^*}[\mathbf{y}]$$

which is the sum of the variance from observation noise and the sample covariance from $T$ stochastic forward passes though the network.

**Predictive Log Likelihood**   We use the Predictive Log Likelihood (PLL) as a measure to estimate the model's uncertainty quality. For a certain test point $(\mathbf{y}_i, \mathbf{x}_i)$, the PLL definition and approximation can be expressed as:

$$
\begin{aligned}
\text{PLL}(f_{\boldsymbol{\omega}}(\mathbf{x}), (\mathbf{y}_i, \mathbf{x}_i)) &= \log p(\mathbf{y}_i | f_{\boldsymbol{\omega}}(\mathbf{x}_i)) \\
&= \log \int f_{\boldsymbol{\omega}}(\mathbf{x}_i, \mathbf{y}_i) p(\boldsymbol{\omega}|\mathbf{D})\mathrm{d}\boldsymbol{\omega} \\
&\approx \log \int f_{\boldsymbol{\omega}}(\mathbf{x}_i, \mathbf{y}_i) q_{\boldsymbol{\theta}}(\boldsymbol{\omega})\mathrm{d}\boldsymbol{\omega} \\
&\approx \log \sum_{j=1}^{T} p(\mathbf{y}_i | f_{\hat{\boldsymbol{\omega}}_j}(\mathbf{x}_i))
\end{aligned}
$$

where $\hat{\boldsymbol{\omega}}_j$ represents a sampled set of stochastic parameters from the approximate posterior distrubtion $q_{\boldsymbol{\theta}}(\boldsymbol{\omega})$ and we take a MC integration with $T$ samples. For regression, due to the iid Gaussian noise, we can further develop the derivation into the form we use when sampling:

$$
\begin{aligned}
\text{PLL}(f_{\boldsymbol{\omega}}(\mathbf{x}), (\mathbf{y}_i, \mathbf{x}_i)) &= \log \sum_{i=1}^{T} \mathcal{N}(\mathbf{y}_i | f_{\hat{\boldsymbol{\omega}}_j}(\mathbf{x}_i), \tau^{-1}\mathbf{I}) \\
&= \text{logsumexp}_{j=1,\ldots,T}\big( -\frac{1}{2}||\mathbf{y}_i - f_{\hat{\boldsymbol{\omega}}_j}(\mathbf{x}_i)||^2\big) \\
&\quad + \log T - \frac{1}{2}\log 2\pi + \frac{1}{2}\log \tau
\end{aligned}
$$

Note that PLL makes no assumption on the form of the approximate predictive distribution. The measure is based on repeated sampling $\hat{\boldsymbol{\omega}}_j$ from $q_{\boldsymbol{\theta}}(\boldsymbol{\omega})$, which may be highly multimodal (see section 3.4).

## 6.5 PRIOR

We assume training by SGD with mini-batch size $M$, L2-regularization on weights and Fully Connected layers. With $\theta_k \in \boldsymbol{\theta}$, equivalence between the objectives of Eq. (1) and (2) then requires:

$$
\begin{aligned}
\frac{\partial}{\partial \theta_k} \text{KL}(q_{\boldsymbol{\theta}}(\boldsymbol{\omega})||p(\boldsymbol{\omega})) &= N\tau \frac{\partial}{\partial \theta_k} \Omega(\boldsymbol{\theta}) \\
&= N\tau \frac{\partial}{\partial \theta_k} \sum_{l=1}^{L} \lambda_l ||\mathbf{W}^l||^2
\end{aligned}
\tag{5}
$$

To proceed with the LHS of Eq. (5) we first need to find the approximate posterior $q_{\boldsymbol{\theta}}(\boldsymbol{\omega})$ that batch normalization induces. As shown in Appendix 6.3, with some weak assumptions and approximations the Central Limit Theorem (CLT) yields Gaussian distributions of the stochastic variables $\mu_{\mathbf{B}}^u, \sigma_{\mathbf{B}}^u$, for large enough $M$:

$$
\begin{aligned}
\mu_{\mathbf{B}}^u &\overset{\propto}{\sim} \mathcal{N}(\mu^u, \frac{(\sigma^u)^2}{M}), \\
\sigma_{\mathbf{B}}^u &\overset{\propto}{\sim} \mathcal{N}(\sigma^u, \frac{\mathbb{E}[(h^u - \mu^u)^4] - (\sigma^u)^4}{4(\sigma^u)^2 M})
\end{aligned}
\tag{6}
$$

where $\mu^u$ and $\sigma^u$ are the *population-level* moments (i.e. moments over $\mathbf{D}$), and $h^u$ is the BN unit's input. We use $i$ as an index of the set of stochastic variables, i.e. $\omega_i \in \{\boldsymbol{\mu}_{\mathbf{B}}^{1:L}, \boldsymbol{\sigma}_{\mathbf{B}}^{1:L}\}$, and denote by $\omega_i^l$ the stochastic variables in a certain layer, $\omega_i^l \in \{\boldsymbol{\mu}_{\mathbf{B}}^l, \boldsymbol{\sigma}_{\mathbf{B}}^l\}$. We assume $q_{\boldsymbol{\theta}}(\boldsymbol{\omega})$ and $p(\boldsymbol{\omega})$ factorize over all individual $\omega_i$, i.e. independence between all stochastic variables.[11] As shown in Eq. (3) in Appendix 6.2, the factorized distributions yield:

$$
\text{KL}(q_{\boldsymbol{\theta}}(\boldsymbol{\omega})||p(\boldsymbol{\omega})) = \sum_i \text{KL}(q_{\boldsymbol{\theta}}(\omega_i)||p(\omega_i))
$$

Note that each BN unit produces two $\text{KL}(q_{\boldsymbol{\theta}}(\omega_i)||p(\omega_i))$ terms: one for $\omega_i = \mu_{\mathbf{B}}^u$ and one for $\omega_i = \sigma_{\mathbf{B}}^u$.

We assume a Gaussian prior $p(\omega_i) = \mathcal{N}(\mu_p, \sigma_p^2)$ and, for consistency, use the notation $q_{\boldsymbol{\theta}}(\omega_i) = \mathcal{N}(\mu_q, \sigma_q^2)$. As shown in Eq. (4) in Appendix 6.2:

$$
\text{KL}(q_{\boldsymbol{\theta}}(\omega_i)||p(\omega_i)) = \ln \frac{\sigma_p}{\sigma_q} + \frac{\sigma_q^2 + (\mu_q - \mu_p)^2}{2\sigma_p^2} - \frac{1}{2}
$$

Then, letting $(\cdot)'$ denote $\frac{\partial}{\partial \theta_k}(\cdot)$:

$$
\begin{aligned}
\frac{\partial}{\partial \theta_k} \text{KL}(q_{\boldsymbol{\theta}}(\omega_i)||p(\omega_i)) &= \frac{2\sigma_q \sigma_p \sigma_q' - \sigma_p'(2\sigma_q^2 - (\mu_q - \mu_p)^2)}{\sigma_p^3} + \frac{(\mu_q - \mu_p)(\mu_q' - \mu_p')}{\sigma_p^2} \\
&= \frac{2\sigma_p \sigma_q \sigma_q'}{\sigma_p^3} + \frac{(\mu_q - \mu_p)\mu_q'}{\sigma_p^2}
\end{aligned}
\tag{7}
$$

where the last step makes use of the fact that $\mu_p' = 0$ and $\sigma_p' = 0$ (as $p(\omega_i)$ cannot depend on $\theta_k$, which changes during training).

We assume that only parameters preceding a BN unit in the same layer affects the unit's stochastic parameters, such that the stochastic variables in the $j$:th BN unit are only affected by weights in $\theta_k \in \mathbf{W}^{l,(j)}$. Let the vector of average inputs over $\mathbf{D}$ from the preceding layers be denoted by $\bar{x}$. We denote a weight connecting the $m$:th input unit to the $j$:th BN unit by $\mathbf{W}^{(j,m)}$. For such weights,

---

[11]The empirical distributions have been numerically checked to be linearly independent and the joint distribution is close to a bi-variate Gaussian.

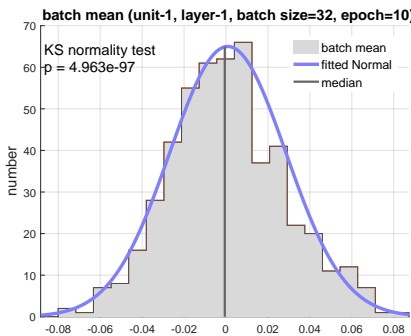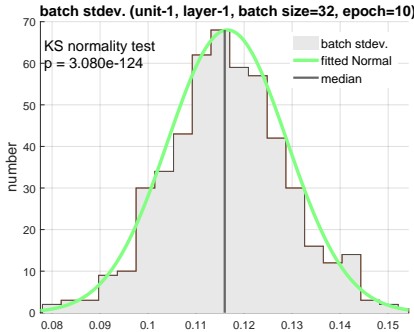

Figure 4: Batch statistics used to train the network are normal. A one-sample Kolmogorov-Smirnov test checks that $\mu_{\mathbf{B}}$ and $\sigma_{\mathbf{B}}$ come from a standard normal distribution. More examples are available in Appendix 6.7.

we need to derive $\mu'_q$ and $\sigma'_q$, for both $\omega_i = \mu_{\mathbf{B}}^u$ and $\omega_i = \sigma_{\mathbf{B}}^u$. Starting with $\omega_i = \mu_{\mathbf{B}}^u$:

$$\mu'_q = \frac{\partial}{\partial \mathbf{W}^{(j,m)}} \frac{\sum_{\boldsymbol{x} \in \mathbf{D}} \mathbf{W}^{(j)} \boldsymbol{x}}{N} = \bar{\boldsymbol{x}}^{(m)}$$

$$\sigma'_q = \frac{\partial}{\partial \mathbf{W}^{(j,m)}} \left[ \frac{\sum_{\boldsymbol{x} \in \mathbf{D}} (\mathbf{W}^{(j)} \boldsymbol{x} - \mu_q)^2}{NM} \right]^{\frac{1}{2}}$$

$$= \frac{1}{2} \left[ \frac{\sum_{\boldsymbol{x} \in \mathbf{D}} (\mathbf{W}^{(j)} \boldsymbol{x} - \mu_q)^2}{NM} \right]^{-\frac{1}{2}} \frac{\sum_{\boldsymbol{x} \in \mathbf{D}} 2(\mathbf{W}^{(j)} \boldsymbol{x} - \mu_q)(\boldsymbol{x}^{(m)} - \bar{\boldsymbol{x}}^{(m)})}{NM} = 0$$

Using the result that $\sigma'_q = 0$, one can easily find for $\omega_i = \sigma_{\mathbf{B}}^u$ that $\mu'_q = 0$ and $\sigma'_q = 0$, nullifying Eq. (7). We need only consider the partial derivatives of the KL divergence terms where $\omega_i = \mu_{\mathbf{B}}^u$. If we let $\mu_p = 0$, Eq. (7) reduces to:

$$\frac{\partial}{\partial \mathbf{W}^{(j,m)}} \mathrm{KL}(q_{\boldsymbol{\theta}}(\omega_i) || p(\omega_i)) = \frac{\mu_q \bar{\boldsymbol{x}}^{(m)}}{\sigma_p^2} = \frac{\bar{\boldsymbol{x}}^{(m)} \mathbf{W}^{(j)} \bar{\boldsymbol{x}}}{\sigma_p^2}$$

for each BN unit and connecting weights from the previous layer, $\mathbf{W}^{(j,m)}$. Taking partial derivatives for all $\mathrm{KL}(q_{\boldsymbol{\theta}}(\omega_i) || p(\omega_i))$ components in the layer:

$$\sum_{\omega_i^l} \sum_j \sum_m \frac{\partial}{\partial \mathbf{W}^{(j,m)}} \mathrm{KL}(q_{\boldsymbol{\theta}}(\omega_i) || p(\omega_i)) = \sum_m \left[ \bar{\boldsymbol{x}}^{(m)} \right] * \sum_j \sum_m \frac{\mathbf{W}^{(j,m)} \bar{\boldsymbol{x}}^{(m)}}{\sigma_p^2} \qquad (8)$$

We consider ReLU activations, such that for large $N$, $\bar{\boldsymbol{x}}^{(m)} > 0 \quad \forall m$. Note that this does not hold for the first layer (which is possibly normalized), but the effect of including these weights in the L2-regularization would be smaller the deeper the network. We assume most outputs from previous layer's BN units remain normalized through the scale and shift transformation, such that we can approximate $\bar{\boldsymbol{x}}^{(m)}$ by the average of all input units over test data, $x$ in Eq. (8). With $J_{l-1}$ units in the input layer, setting $\sigma_p^2 = \frac{J_{l-1} x^2}{2N\lambda_l}$, such that $p(\mu_{\mathbf{B}}^u) = \mathcal{N}(0, \frac{J_{l-1} x^2}{2N\tau\lambda_l})$ would then reconcile Eq. (5), for any Gaussian $p(\sigma_{\mathbf{B}}^u)$.

## 6.6 EXTENDED EXPERIMENTAL RESULTS

Below, we provide extended results measuring uncertainty quality. In Tables 3 and 4, we provide tables showing the mean $\overline{\mathrm{CRPS}}$ and $\overline{\mathrm{PLL}}$ values for MCBN and MCDO. These results indicate that MCBN performs on par or better than MCDO across several datasets. In Table 5 we provide the raw PLL and CRPS results for MCBN and MCDO. In Table 6 we provide RMSE results of the MCBN and MCDO networks in comparison with non-stochastic BN and DO networks. These results indicate that the procedure of multiple forward passes in MCBN and MCDO show slight improvements in the accuracy of the network.

In Figure 5 and Figure 6, we provide a full set of our uncertainty quality visualization plots, where errors in predictions are sorted by estimated uncertainty. The shaded areas show the model uncertainty and gray dots show absolute prediction errors on the test set. A gray line depicts a running

mean of the errors. The dashed line indicates the optimized constant uncertainty. In these plots, we can see a correlation between estimated uncertainty (shaded area) and mean error (gray). This trend indicates that the model uncertainty estimates can recognize samples with larger (or smaller) potential for predictive errors.

| Dataset | $\overline{\text{CRPS}}$ | | | |
|---|---|---|---|---|
| | MCBN | $p$-value | MCDO | $p$-value |
| Boston Housing | **8.50** ±0.86 | 6.39e-10 | 3.06 ±0.33 | 1.64e-9 |
| Concrete | **3.91** ±0.25 | 4.53e-14 | 0.93 ±0.41 | 3.13e-2 |
| Energy Efficiency | **5.75** ±0.52 | 6.71e-11 | 1.37 ±0.89 | 1.38e-1 |
| Kinematics 8nm | **2.85** ±0.18 | 2.33e-14 | 1.82 ±0.14 | 1.64e-12 |
| Power Plant | **0.24** ±0.05 | 2.32e-4 | -0.44 ±0.05 | 2.17e-8 |
| Protein | **2.66** ±0.10 | 2.77-12 | 0.99 ±0.08 | 2.34e-12 |
| Wine Quality (Red) | 0.26 ±0.07 | 1.26e-3 | **2.00** ±0.21 | 1.83e-9 |
| Yacht Hydrodynamics | -56.39 ±14.27 | 5.94e-4 | **21.42** ±2.99 | 2.16e-7 |

Table 3: $\overline{\text{CRPS}}$ measured on eight datasets over 25 random 80-20 splits of the data. Mean values for MCBN and MCDO are reported along with standard error. A significance test was performed to check if $\overline{\text{CRPS}}$ significantly exceeds the baseline. The $p$-value from a one sample t-test is reported. Marked in bold is the best performing method versus its baseline.

| Dataset | $\overline{\text{PLL}}$ | | | |
|---|---|---|---|---|
| | MCBN | $p$-value | MCDO | $p$-value |
| Boston Housing | **10.49** ±1.35 | 5.41e-8 | 5.51 ±1.05 | 2.20e-5 |
| Concrete | -36.36 ±12.12 | 6.19e-3 | **10.92** ±1.78 | 2.34e-6 |
| Energy Efficiency | **10.89** ±1.16 | 1.79e-9 | -14.28 ±5.15 | 1.06e-2 |
| Kinematics 8nm | **1.68** ±0.37 | 1.29e-4 | -0.26 ±0.18 | 1.53e-1 |
| Power Plant | 0.33 ±0.14 | 2.72e-2 | **3.52** ±0.23 | 1.12e-13 |
| Protein | 2.56 ±0.23 | 4.28e-11 | **6.23** ±0.19 | 2.57e-21 |
| Wine Quality (Red) | 0.19 ±0.09 | 3.72e-2 | **2.91** ±0.35 | 1.84e-8 |
| Yacht Hydrodynamics | **45.58** ±5.18 | 5.67e-9 | -41.54 ±31.37 | 1.97e-1 |

Table 4: $\overline{\text{PLL}}$ measured on eight datasets over 25 random 80-20 splits of the data. Mean values for MCBN and MCDO are reported along with standard error. A significance test was performed to check if $\overline{\text{PLL}}$ significantly exceeds the baseline. The $p$-value from a one sample t-test is reported. Marked in bold is the best performing method versus its baseline.

| Dataset | CRPS | | PLL | |
|---|---|---|---|---|
| | MCBN | MCDO | MCBN | MCDO |
| Boston Housing | 1.45±0.02 | **1.41**±0.02 | -2.38±0.02 | **-2.35**±0.02 |
| Concrete | **2.40**±0.04 | 2.42±0.04 | -3.45±0.11 | **-2.94**±0.02 |
| Energy Efficiency | 0.33±0.01 | **0.26**±0.00 | -0.94±0.04 | **-0.80**±0.04 |
| Kinematics 8nm | 0.04±0.00 | **0.04**±0.00 | **1.21**±0.01 | 1.24±0.00 |
| Power Plant | 2.00±0.01 | **2.00**±0.01 | -2.75±0.00 | **-2.72**±0.01 |
| Protein Tertiary Structure | **1.95**±0.01 | 1.95±0.00 | -2.73±0.00 | **-2.70**±0.00 |
| Wine Quality (Red) | 0.34±0.00 | **0.33**±0.00 | -0.95±0.01 | **-0.89**±0.01 |
| Yacht Hydrodynamics | 0.68±0.02 | **0.32**±0.01 | **-1.39**±0.03 | -2.57±0.69 |

Table 5: CRPS and PLL measured on eight datasets over 25 random 80-20 splits of the data. Mean values and standard errors are reported for MCBN and MCDO. Marked in bold is the best performing method for each metric.

## 6.7 BATCH NORMALIZATION STATISTICS

In Figure 7 and Figure 8, we provide statistics on the batch normalization parameters used for training. The plots show the distribution of BN mean and BN variance over different mini-batches

| Dataset | RMSE | | | |
| --- | --- | --- | --- | --- |
| | MCBN | BN | MCDO | DO |
| Boston Housing | 2.75 ±0.05 | 2.77 ±0.05 | **2.65** ±0.05 | 2.69 ±0.05 |
| Concrete | **4.78** ±0.09 | 4.89 ±0.08 | 4.80 ±0.10 | 4.99 ±0.10 |
| Energy Efficiency | 0.59 ±0.02 | 0.57 ±0.01 | **0.47** ±0.01 | 0.49 ±0.01 |
| Kinematics 8nm | 0.07 ±0.00 | 0.07 ±0.00 | **0.07** ±0.00 | 0.07 ±0.00 |
| Power Plant | 3.74 ±0.01 | 3.74 ±0.01 | 3.74 ±0.02 | **3.72** ±0.02 |
| Protein | 3.66 ±0.01 | 3.69 ±0.01 | **3.66** ±0.01 | 3.68 ±0.01 |
| Wine Quality (Red) | 0.62 ±0.00 | 0.62 ±0.00 | **0.60** ±0.00 | 0.61 ±0.00 |
| Yacht Hydrodynamics | 1.23 ±0.05 | 1.28 ±0.06 | 0.75 ±0.03 | **0.72** ±0.04 |

Table 6: RMSE measured on eight datasets over 25 random 80-20 splits of the data. Mean values and standard errors are reported for MCBN and MCDO as well as conventional non-Bayesian models BN and DO. Marked in bold is the best performing method overall.

of an actual training of Yacht dataset for one unit in the first hidden layer and the second hidden layer. Data is provided for different epochs and for different batch sizes.

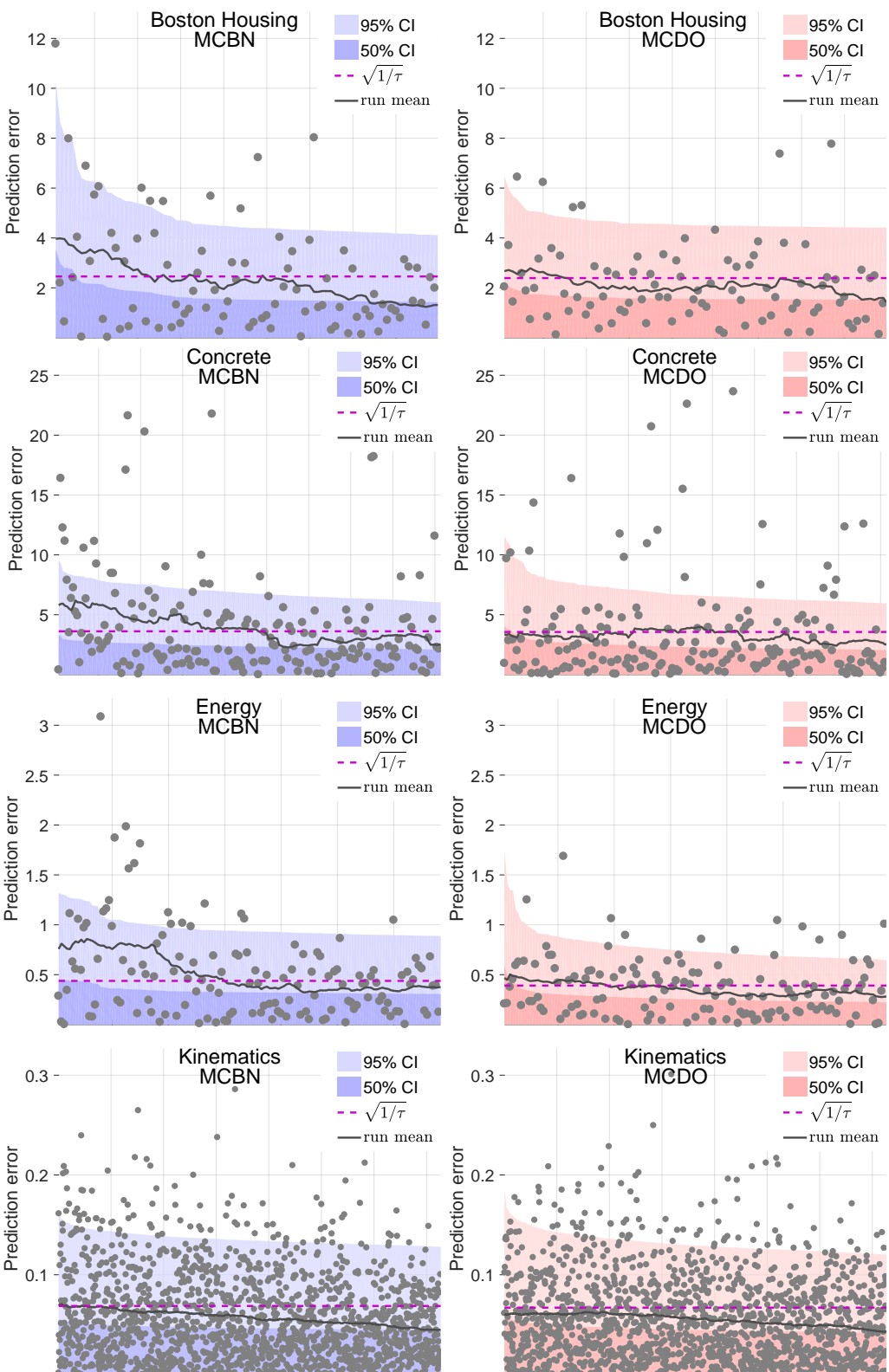

Figure 5: Errors in predictions (gray dots) sorted by estimated uncertainty on select datasets. The shaded areas show MCBN's (blue) and MCDO's (red) model uncertainty (light area 95% CI, dark area 50% CI). Gray dots show absolute prediction errors on the test set, and the gray line depicts a running mean of the errors. The dashed line indicates the optimized constant uncertainty. A correlation between estimated uncertainty (shaded area) and mean error (gray) indicates the uncertainty estimates are meaningful for estimating errors.

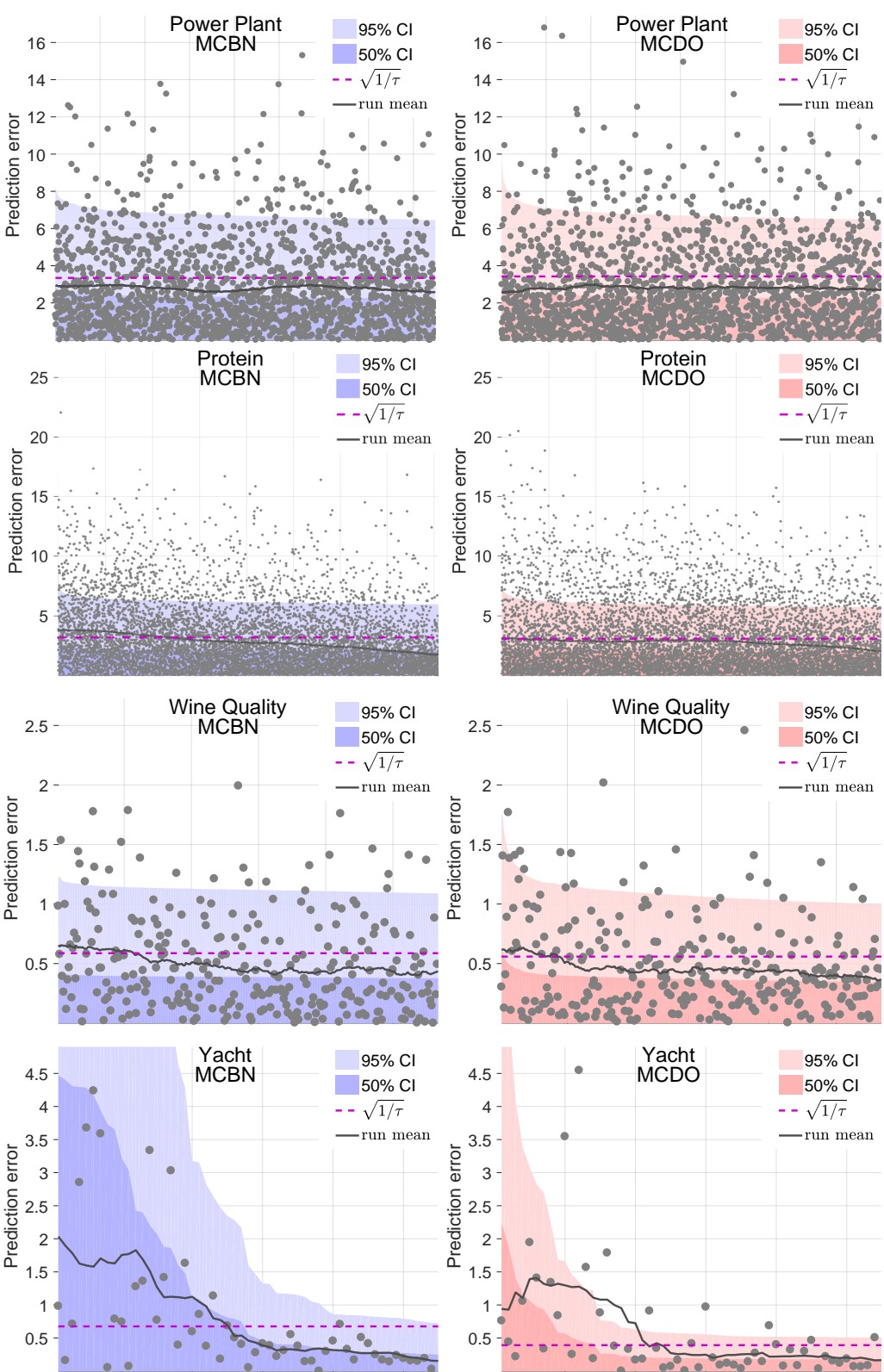

Figure 6: Errors in predictions (gray dots) sorted by estimated uncertainty on select datasets. The shaded areas show MCBN's (blue) and MCDO's (red) model uncertainty (light area 95% CI, dark area 50% CI). Gray dots show absolute prediction errors on the test set, and the gray line depicts a running mean of the errors. The dashed line indicates the optimized constant uncertainty. A correlation between estimated uncertainty (shaded area) and mean error (gray) indicates the uncertainty estimates are meaningful for estimating errors.

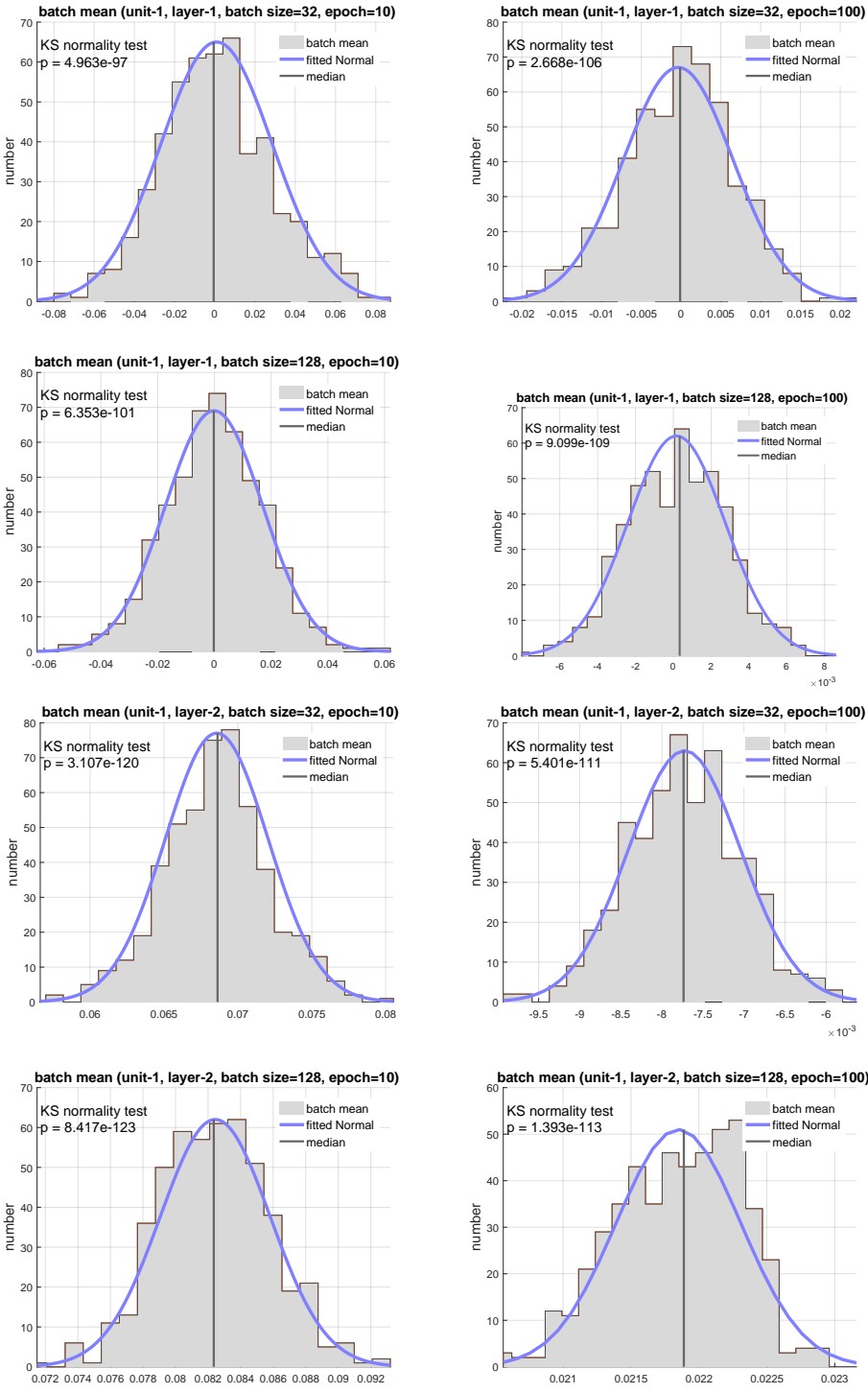

Figure 7: **The distribution of means** of mini-batches during training of one of our datasets. The distribution closely follows our analytically approximated Gaussian distribution. The data is collected for one unit of each layer and is provided for different epochs and for different batch sizes.

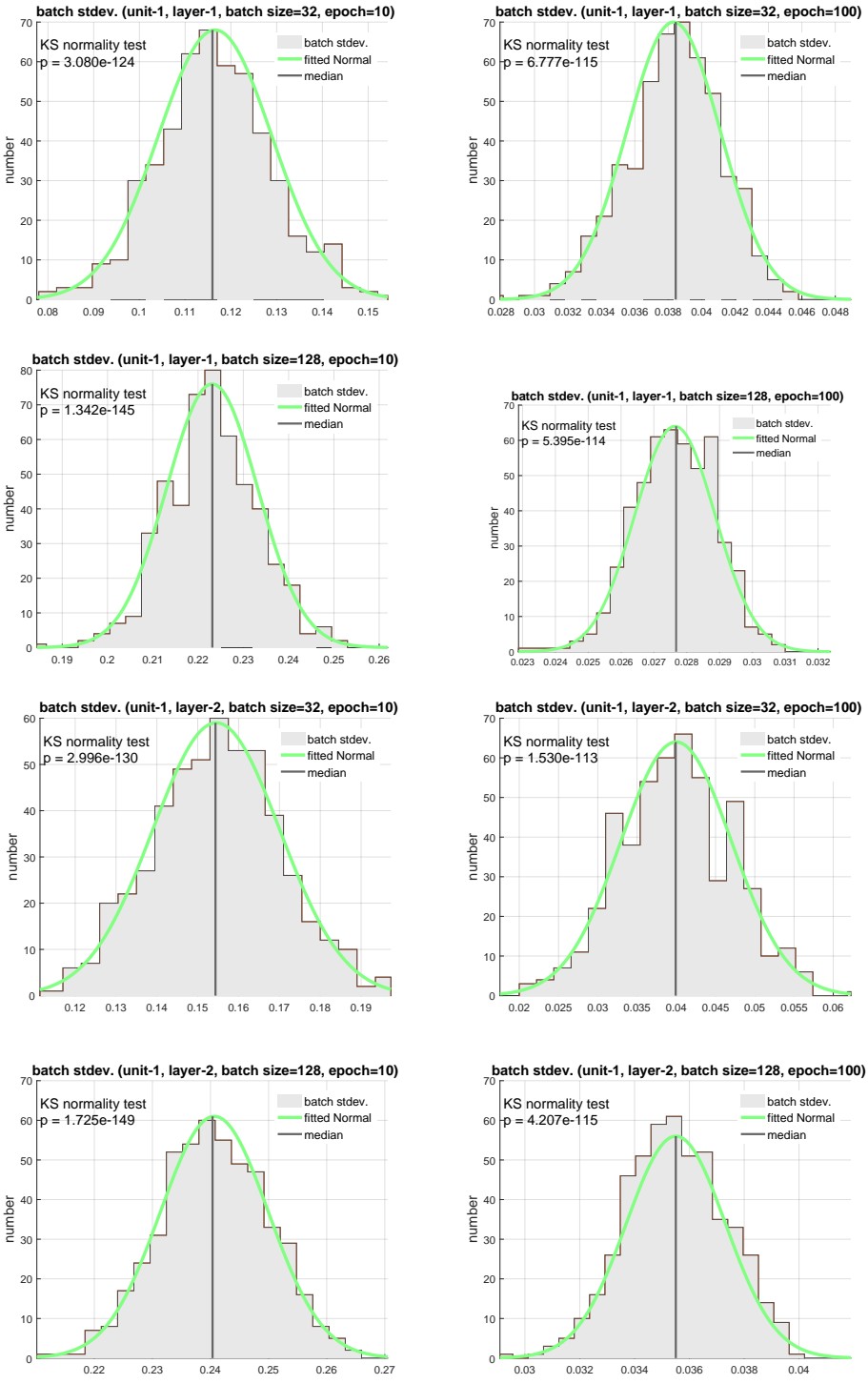

Figure 8: **The distribution of standard deviation** of mini-batches during training of one of our datasets. The distribution closely follows our analytically approximated Gaussian distribution. The data is collected for one unit of each layer and is provided for different epochs and for different batch sizes.

