# OpenReview forum: "Bayesian Uncertainty Estimation for Batch Normalized Deep Networks"
_ICLR.cc/2018/Conference — Reject_

### Official Review · AnonReviewer2 · 2017-11-17
**A dense paper with a few key open questions**

**Rating:** 5
**Confidence:** 3

**Review:**

This paper proposes an approximate method to construct Bayesian uncertainty estimates in networks trained with batch normalization.

There is a lot going on in this paper. Although the overall presentation is clean, there are few key shortfalls (see below). Overall, the reported functionality is nice, although the experimental results are difficult to intepret (despite laudable effort by the authors to make them intuitive).

Some open questions that I find crucial:

* How exactly is the “stochastic forward-pass” performed that gives rise to the moment estimates? This step is the real meat of the paper, yet I struggle to find a concrete definition in the text. Is this really just an average over a few recent weights during optimization? If so, how is this method specific to batch normalization? Maybe I’m showing my own lack of understanding here, but it’s worrying that the actual sampling technique is not explained anywhere. This relates to a larger point about the paper's main point: What, exactly, is the Bayesian interpretation of batch normalization proposed here? In Bayesian Dropout, there is an explicit variational objective. Here, this is replaced by an implicit regularizer. The argument in Section 3.3 seems rather weak to me. To paraphrase it: If the prior vanishes, so does the regularizer. Fine. But what's the regularizer that's vanishing? The sentence that "the influence of the prior diminishes as the size of the training data increases" is debatable for something as over-parametrized as a DNN. I wouldn't be surprised that there are many directions in the weight-space of a trained DNN along which the posterior is dominated by the prior.

* I’m confused about the statements made about the “constant uncertainty” baseline. First off, how is this (constant) width of the predictive region chosen? Did I miss this, or is it not explained anywhere? Unless I misunderstand the definition of CRPS and PLL, that width should matter, no? Then, the paragraph at the end of page 8 is worrying: The authors essentially say that the constant baseline is quite close to the estimate constructed in their work because constant uncertainty is “quite a reasonable baseline”. That can hardly be true (if it is, then it puts the entire paper into question! If trivial uncertainty is almost as good as this method, isn't the method trivial, too?).
On a related point: What would Figure 2 look like for the constand uncertainty setting? Just a horizontal line in blue and red? But at which level?

I like this paper. It is presented well (modulo the above problems), and it makes some strong points. But I’m worried about the empirical evaluation, and the omission of crucial algorithmic details. They may hide serious problems.

---

> ### Author Response · Authors · 2018-01-03
> **Response**
>
> Thank you for your comments. We hope the to answer your concerns below.
>
> We have clarified the description on how MCBN is used in section (3.4). Below we first describe how the model is used, then discuss the rationale.
>
> The network is trained as a regular BN model. The difference is in using the model for prediction. We estimate the mean and variance of the predictive distribution at a new input x by MC sampling:
>
> For a total of T times:
> - Sample a batch B from the training data D (with the same batch size M that was used during training).
> - Update the BN units’ means and variances with B. This corresponds to sampling from the approximate predictive distribution q_theta(omega).
> - Perform a forward pass to get the output y_t with this particular sample of omega.
>
> From the T output samples y_t we estimate:
> - The mean of the predictive distribution as the sample mean of y.
> - The variance of the predictive distribution (for regression) as sum of the sample variance of y and variance from constant observation noise, tau^-1*I.
>
> Derivations of these estimates are given in Appendix 6.4.
>
> These moments do not disclose the form of the approximate posterior distribution p*. It is likely multimodal, but we have added a proof in section (3.4) that it can be approximated by a Gaussian for each output dimension. We may therefore fit a Gaussian distribution to the estimated moments as an estimate of the predictive distribution of new input x.
>
> What is the Bayesian interpretation of batch normalization?
> From a Bayesian perspective, sampling a batch and updating the stochastic parameters omega (all BN units’ mean and std dev. parameters) during training means that the trained network is equivalent to having minimized the KL divergence of KL(approximate posterior || true posterior) wrt theta. Therefore q_theta(omega) (the joint distribution of the network’s stochastic parameters) is an approximation of the true posterior, restricted to lie within the domain of our parametric network, and source of randomness (sampling batches of size M from D). q_theta(omega) is  an approximation of the true posterior under these restrictions, and by the limitations intrinsic to KL divergence minimization. The definition of q_theta(omega) has been clarified in section (3.2), and its equivalence to KL divergence minimization is discussed in section (3.4).
>
> It is correct that q_theta(omega) is defined implicitly, by our network architecture but also M and D. Note that the approximate posterior q_theta(omega) must be consistent during and after training. This means that the mini-batch size M and the dataset from which B is sampled (i.e. the training data D) must be kept after training when taking omega samples for estimating the predictive distribution. (we have not evaluated our modeled Gaussian approximation from Appendix 6.3)
>
> We agree that dropping the regularizer/prior is hard to motivate from a Bayesian perspective. We have removed this discussion. We now model an approximate prior in Appendix 6.5. Our implied prior over batch means is p(mu) = N(0, (J * x_bar^2) / (2 * N * tau * lambda)). From a VA perspective, too strong a regularization for a given dataset size could be seen as constraining the prior distribution of BN units’ means, effectively narrowing the approximate posterior.
>
> Evaluation baselines
> We evaluate MCBN and MCDO using two standard metrics of predictive distribution quality: PLL and CRPS. It is difficult though to directly compare different models based on these metrics alone unless the models produce the same means at every test point (which does not happen in practice). If we were to compare MCBN to MCDO and find that e.g. PLL was in MCBN’s favor, we would not be able to say whether the predictive distribution of MCBN makes sense or not – the outperformance could simply be a result of BN fitting the model better to the data.
>
> We normalize the measures with an upper- and lower bound. CUBN and CUDO represent the lower bound. These models produce the same means as MCBN and MCDO respectively, but always estimate a constant (validation-optimized) variance. This is the best we can do, if we were to always assume the same predictive variance. Any improvement indicates that the MC models estimate uncertainty in a sensible way. This has been clarified in section 4.2
>
> The upper bound also produce the same target estimates, but the predicted varianceoptimizes CRPS and PLL respectively, for each test data point. This is the best-case scenario - any change for a single test data point would yield a lower score. By normalizing the scores achieved by MCBN and MCDO between these bounds, we not only verify that the models are better than the constant uncertainty baselines (i.e. model input-dependent variance sensibly), but also achieve an estimate of how close the modeled variance is to the absolute best case.
>
> In Figure 2, we have included the CU- models’ constant uncertainty as one standard deviation, given by the dashed line.

---

### Official Review · AnonReviewer3 · 2017-11-27
**Using Batch norm at test time to obtain uncertainty estimate**

**Rating:** 5
**Confidence:** 4

**Review:**

*Summary*

The paper proposes using batch normalisation at test time to get the predictive uncertainty. The stochasticity of the prediction comes from different minibatches of training data that were used to normalise the activity/pre-activation values at each layer. This is justified by an argument that using batch norm is doing variational inference, so one should use the approximate posterior provided by batch norm at prediction time. Several experiments show Monte Carlo prediction at test time using batch norm is better than dropout.

*Originality and significance*

As far as I understand, almost learning algorithms similar to equation 2 can be recast as variational inference under equation 1. However, the critical questions are what is the corresponding prior, what is the approximating density, what are the additional approximations to obtain 2, and whether the approximation is a good approximation for getting closer to the posterior/obtain better prediction.

It is not clear to me from the presentation what the q(w) density is -- whether this is explicit (as in vanilla Gaussian VI or MC dropout), or implicit (the stochasticity on the activity h due to batch norm induces an equivalence q on w).

From a Bayesian perspective, it is also not satisfying to ignore the regularisation term by an empirical heuristic provided in the batch norm paper [small \lambda] -- what is the rationale of this? Can this be explained by comparing the variational free-energy.

The experiments also do not compare to modern variational inference methods using the reparameterisation trick with Gaussian variational approximations (see Blundell et al 2016) or richer variational families (see e.g. Louizos and Welling, 2016, 2017). The VI method included in the PBP paper (Hernandez-Lobato and Adams, 2015) does not use the reparameterisation trick, which has been found to reduce variance and improve over Graves' VI method.

*Clarity*
The paper is in general well written and easy to understand.

*Additional comments*

Page 2: Monte Carlo Droput --> Dropout
Page 3 related work: (Adams, 2015) should be (Hernandez-Lobato and Adams, 2015)

---

> ### Author Response · Authors · 2018-01-03
> **Response**
>
> Thank you for your comments. We hope to address your concerns below.
>
> We have added derivations of the implied prior for networks with L2-regularization (summarized in Section 3.3 and fully derived in Appendix 6.5). The derivations assume fully connected layers with ReLU activations as used in most modern batch-normalized networks. We use the modeled approximate posterior q_theta(omega) from Appendix 6.3. We assumes a factorized Gaussian distribution over all stochastic variables, and that only parameters in the current layer affects the distribution of its stochastic variables.
>
> The implied prior on BN units’ std. dev. terms are Gaussian, with arbitrary moments.
> The implied prior on all BN units’ means for each layer are:
>
> p(mu) = N(0, (J * x_bar^2) / (2 * N * tau * lambda))
>
> J: n.o. input units to the layer
> x_bar: average input from all input units, across training data D
> N: Size of training dataset
> tau: inverse variance from constant observation noise
> lambda: the layer’s L2 regularization coefficient
>
> This prior is an approximation, and is only accurate if the average input for each input unit over D is identical (which is the case if the scale and shift transformation is identical for all units). In the absence of scale and shift transformations from the previous BN layer, it converges towards an exact prior for large training datasets and deep networks (under the assumptions of q_theta(omega) and the factorized Gaussian).
>
> With this implied prior, strong regularization corresponds to a prior over BN unit means with small variance. From a VA perspective, too strong a regularization for a given dataset size could be seen as constraining the prior distribution of BN units’ means, effectively narrowing the approximate posterior.
>
> What exactly is the Bayesian interpretation of batch normalization proposed here (and what is the density q)?
> From a Bayesian perspective, sampling a batch and updating the stochastic parameters (all BN units’ mean and std dev. parameters) during training means that the trained network is equivalent to having minimized the KL divergence of KL(approximate posterior || true posterior) wrt theta. Therefore q_theta(omega) (the joint distribution of the network’s stochastic parameters) is an approx. of the true posterior, restricted to lie within the domain of our parametric network, and source of randomness (sampling batches of size M from D). q_theta(omega) is an approximation of the true posterior under these restrictions, and by the limitations intrinsic to KL divergence minimization. The definition of q_theta(omega) has been clarified in section (3.2), and its equivalence to KL divergence minimization is discussed in section (3.4).
>
> It is correct that q_theta(omega) is defined implicitly, by our network architecture but also M and D. This means that the approximate posterior q_theta(omega) must be consistent during and after training. This means that the mini-batch size M and the dataset from which B is sampled (i.e. the training data D) must be kept after training when taking omega samples for estimating the predictive distribution. Alternatively, one could use our modeled q_theta(omega) as factorized Gaussians - but we leave this as suggestions for future research.
>
> What are the approximations to obtain the approximate posterior, and is our approximation close to the true posterior?
> The modeling of q_theta(omega) from BN as Gaussian over all the network’s stochastic parameters is an approximation that by CLT relies on a large enough n.o. input units, as shown in Appendix (6.3). We additionally assume that this factorizes over all individual stochastic parameters, for the derivations of the implied prior in Appendix 6.5. How suitable this simplification of q_theta(omega) is for sampling in the predictive distribution is difficult to say without evaluating the quality of the predictive distribution empirically. However, the modeling allows us to study the implied prior, which would be difficult with the random variable as a selection of mini-batch members.
>
> Results comparison to other models
> We have adapted Louizos & Welling’s implementation of Multiplicative Normalizing Flows for Variational Bayesian Neural Network (MNF) for our evaluation. With this we are able to compare our results with a model highly capable of producing complex approximate posteriors. we have included results for three finished datasets in Table 2, and will be continuing to update the results as evaluations finish. So far, the normalized scores are in line with what we observe for MCBN and MCDO - less than 10% for Boston and Concrete, and inconsistent between the metrics for Yacht. The evaluation is performed the same way as for MCBN and MCDO with the exception of the initial grid search hyperparameter selection - we will make sure to apply proper hyperparameter selection to MNF for the camera-ready version of the paper.
>
> We have also corrected the typo on Dropout, and the erroneous reference.

---

### Official Review · AnonReviewer1 · 2017-11-27
**Interesting and relevant but lack of details on prior**

**Rating:** 6
**Confidence:** 4

**Review:**

The authors show how the regularization procedure called batch normalization,
currently being used by most deep learning systems, can be understood as
performing approximate Bayesian inference. The authors compare this approach to
Monte Carlo dropout (another regularization technique which can also be
considered to perform approximate Bayesian inference). The experiments
performed show that the Bayesian view of batch normalization performs similarly
as MC dropout in terms of the estimates of uncertainty that it produces.

Quality:

I found the quality to be low in some aspects. First, the description of what
is the prior used by batch normalization in section 3.3 is unsatisfactory. The
authors basically refer to Appendix 6.4 for the case in which the weight decay
penalty is not zero. The details in that Appendix are almost none, they just
say "it is thus possible to derive the prior...".

The results in Table 2 are a bit confusing. The authors should highlight in
bold face the results of the best performing method.

The authors indicate that they do not need to compare to variational methods
because Gal and Ghahramani 2015 compare already to those methods. However, Gal
and Ghahramani's code used Bayesian optimization methods to tune
hyper-parameters and this code contains a bug that optimizes hyper-parameters
by maximizing performance on the test data. In particular for hyperparameter
selection, they average performance across (subsets of) 5 of the training sets
from the 20x train/test split, and then using the tau which got the best
average performance for all of 20x train/test splits to evaluate performance:

https://github.com/yaringal/DropoutUncertaintyExps/blob/master/bostonHousing/net/experiment_BO.py#L54

Therefore, the claim that

"Since we have established that MCBN performs on par with MCDO, by proxy we
might conclude that MCBN outperforms those VI methods as well."

is not valid.

At the beginning of section 4.3 the authors indicate that they follow in their
experiments the setup of Gal and Ghahramani (2015). However, Gal and Ghahramani
(2015) actually follow Hernández-Lobato and Adams, 2015 so the correct
reference should be the latter one.

Clarity:

The paper is clearly written and easy to follow and understand.

I found confusing how to use the proposed method to obtain estimates of
uncertainty for a particular test data point x_star. The paragraph just above
section 4 says that the authors sample a batch of training data for this, but
assume that the test point x_star has to be included in this batch.
How is this actually done in practice?

Originality:

The proposed contribution is original. This is the first time that a Bayesian
interpretation has been given to the batch normalization regularization
proposal.

Significance:

The paper's contributions are significant. Batch normalization is a very
popular regularization technique and showing that it can be used to obtain
estimates of uncertainty is relevant and significant. Many existing deep
learning systems can use this to produce estimates of uncertainty in their
predictions.

---

> ### Author Response · Authors · 2018-01-03
> **Response**
>
> Thank you for your comments. We appreciate the feedback and address your concerns below.
>
> We have clarified the description on how MCBN is used in section (3.4). Below we first describe how the model is used, then discuss the rationale.
>
> The network is trained as a regular BN model. The difference is in using the model for prediction. We estimate the mean and variance of the predictive distribution at a new input x by MC sampling:
>
> For a total of T times:
> - Sample a batch B from the training data D (with the same batch size M that was used during training).
> - Update the BN units’ means and variances with B. This corresponds to sampling from the approximate predictive distribution q_theta(omega).
> - Perform a forward pass to get the output y_t with this particular sample of omega.
>
> From the T output samples y_t we estimate:
> - The mean of the predictive distribution as the sample mean of y.
> - The variance of the predictive distribution (for regression) as sum of the sample variance of y and variance from constant observation noise, tau^-1*I.
>
> Derivations of these estimates are given in Appendix 6.4.
>
> Note that these moments do not disclose any information about the form of the approximate posterior distribution p*. It is likely multimodal, but we have added a proof in section (3.4) that it can be approximated by a Gaussian for each output dimension (similar to Wang & Manning’s motivation for a Gaussian approximation of Dropout in Fast dropout training). We may therefore fit a Gaussian distribution to the estimated moments as an estimate of the predictive distribution of new input x.
>
> What is the implied prior?
> We have added derivations of the implied prior for networks with L2-regularization (summarized in Section 3.3 and fully derived in Appendix 6.5). The derivations assume fully connected layers with ReLU activations as used in most modern batch-normalized networks. We use the modeled approximate posterior q_theta(omega) from Appendix 6.3. We assumes a factorized Gaussian distribution over all stochastic variables, and that only parameters in the current layer affects the distribution of its stochastic variables.
>
> The implied prior on BN units’ std. dev. terms are Gaussian, with arbitrary moments.
> The implied prior on all BN units’ means for each layer are:
>
> p(mu) = N(0, (J * x_bar^2) / (2 * N * tau * lambda))
>
> J: n.o. input units to the layer
> x_bar: average input from all input units, across training data D
> N: Size of training dataset
> tau: inverse variance from constant observation noise
> lambda: the layer’s L2 regularization coefficient
>
> This prior is an approximation, and is only accurate if the average input for each input unit over D is identical (which is the case if the scale and shift transformation is identical for all units). In the absence of scale and shift transformations from the previous BN layer, it converges towards an exact prior for large training datasets and deep networks (under the assumptions of q_theta(omega) and the factorized Gaussian).
>
> Results comparison to other models
> We have removed claims of proxy comparison. Instead, we have adapted Louizos & Welling’s implementation of Multiplicative Normalizing Flows for Variational Bayesian Neural Network (MNF) for our evaluation. With this we are able to compare our results with a model highly capable of producing complex approximate posteriors. we have included results for three finished datasets in Table 2, and will be continuing to update the results as evaluations finish. So far, the normalized scores are in line with what we observe for MCBN and MCDO - less than 10% for Boston and Concrete, and inconsistent between the metrics for Yacht. The evaluation is performed the same way as for MCBN and MCDO with the exception of the initial grid search hyperparameter selection - we will make sure to apply proper hyperparameter selection to MNF for the camera-ready version of the paper.
>
> Other comments
> Regarding the tables, we have marked in bold the model that performs best relative to its constant uncertainty baseline in Table 2, as well as in Appendix 6.6 Table 3 and 4. In Table 5 (RMSE) we have marked in bold the best performing model overall. We have also corrected the reference regarding the experiment setup to Hernandez-Lobato & Adams (2015)

---

### Author Response · Authors · 2018-01-03
**Response**

We would like to thank the reviewers for their detailed comments and clear questions. Their feedback helped us improve the quality of the paper.

Before addressing individual comments, we would like to reiterate the contributions of this work which we feel are significant and of broad interest to the ML community:
1) Treat batch normalization as a stochastic regularization and thereby consider a batch-normalized network training procedure as approximate Bayesian modeling.
2) Extensive empirical evidence for the efficacy of obtained predictive uncertainty from such a perspective on batch-normalized networks.
3) Analytical study of the induced prior of the stochastic variables.
4) Novel quantitative and qualitative evaluation of the predictive uncertainties.

Considering the fact that nearly all modern networks use batch normalization, our proposed method is of broad interest as it opens the door to uncertainty estimation in existing conventional networks without modifying the network or the training procedure.

Our response to the reviewer comments appear below. We have thoroughly attended to **all** the raised issues. Also, the manuscript has been revised to include additional studies and explanations as requested by the reviewers; most notably an analytical study of the prior and additional experiments.

We had originally addressed all questions in one response, but this far exceeded the character limitation. We will address individual reviewers below. In the interest of swift response now that we have uploaded a revised paper there will be some repetition, we hope you don’t mind this.

---

> ### Author Response · Authors · 2018-01-05
> **Additional results**
>
> We have updated the paper with:
> - added normalized CRPS and PLL results from MNF (Louizos & Welling) on three additional datasets (Table 2 in Section 4.4). These results are in line with what we have observed for MCBN and MCDO.
> - Raw (non-normalized) PLL and CRPS results for MCBN and MCDO (Table 5 in Appendix 6.6)
>
> As we have mentioned in our responses, please note that the evaluation of MNF is performed the same way as for MCBN and MCDO, with the exception of the initial grid search hyperparameter selection. For the camera-ready version of the paper we will make sure to apply proper hyperparameter selection to MNF as well.

---

### Decision · Program_Chairs · 2018-01-29
**ICLR 2018 Conference Acceptance Decision**

**Decision:**

Reject

**Comment:**

This paper shows that batch normalization can be cast as approximate inference in deep neural networks.  This is an appealing result as batch normalization is used in practice in a wide variety of models.   The reviewers found the paper well written and easy to understand and were motivated by underlying idea.  However, they found the empirical analysis lacking and found that there was not enough detail in the main text to verify whether the claims were true.

The authors empirically compared to a recent method showing that dropout can be cast as approximate inference with the claim that by transitivity they were comparing to a variety of recent methods.  AnonReviewer1 casts significant doubt on the results of that work.  This is very unfortunate and not the fault of the authors of this paper.  The authors have since gone to great length to compare to Louizos and Welling, 2017.  Unfortunately, that comparison doesn't appear to be complete in the manuscript.

The main text was also lacking specific detail relating to fundamental parts of the proposed method (noted by all reviewers).

Overall, this paper seems to be tremendously promising and the underlying idea potentially very impactful.  However, given the reviews, it doesn't seem that this paper would achieve its potential impact.  The response from the authors is appreciated and goes a long way to improving the paper.  Taking the reviews into account, adding specific detail about the methodology and model (e.g. the prior) and completing careful empirical analysis will make this a strong paper that should be much more impactful.

---

> ### Author Response · Authors · 2018-02-21
> **Post-decision response from the authors**
>
> We thank the program committee for their time and effort in a demanding review process.
>
> We appreciate the sentiment in the decision from the Program Chairs noting that "this paper seems to be tremendously promising and the underlying idea potentially very impactful", and that "this is an appealing result as batch normalization is used in practice in a wide variety of models. The reviewers found the paper well written and easy to understand and were motivated by underlying idea."
>
> In response to the reviewer comments we made substantial changes to the manuscript during the rebuttal period. We provided a thorough response to the issues raised by the reviewers and uploaded a revision addressing these shortcomings, including a more detailed explanation of the prior and a comparison to MNF. Unfortunately, there was no response from the reviewers to our revision or our comments.
>
> We understand the decision in the context that our changes constitute a substantial change to the original work, and that resources for reviewing are limited. However, we would like to state for the record that we believe our response and revised manuscript (which addressed the criticisms of the reviewers) were not given full consideration.
>
> Regards,
> the authors